# REEFNET: A LARGE-SCALE, TAXONOMICALLY ENRICHED DATASET AND BENCHMARK FOR HARD CORAL CLASSIFICATION

## ABSTRACT

Coral reefs are rapidly declining due to anthropogenic pressures like climate change, underscoring the urgent need for scalable, automated monitoring. We introduce ReefNet, a large public coral reef image dataset with point-label annotations mapped to the World Register of Marine Species (WoRMS). ReefNet aggregates imagery from 76 curated CoralNet sources and an additional site from Al-Wajh in the Red Sea, totaling $\sim$ 925K genus-level hard coral annotations with expert-verified labels. Unlike prior datasets, often limited by size, geography, or coarse labels and not ML-ready, **ReefNet** offers *fine-grained, taxonomically mapped* labels at a global scale to WoRMS. We propose two evaluation settings: (i) a *within-source* benchmark that partitions each source's images for localized evaluation, and (ii) a *cross-source* benchmark that withholds entire sources to test domain generalization. We analyze both supervised and zero-shot classification performance on ReefNet, and find that while supervised within-source performance is promising, supervised performance drops sharply across domains, and performance is low across the board for zero-shot models, especially for rare and visually similar genera, providing a challenging benchmark intended to catalyze advances in domain generalization and fine-grained coral classification. We will release our dataset, benchmarking code, and pretrained models to advance robust, domain-adaptive, global coral reef monitoring and conservation.

## 1 INTRODUCTION

Coral reefs are among the most biodiverse ecosystems on Earth, providing immense ecological and economic value (Barbier et al., 2011; Spalding et al., 2017). Yet these vibrant habitats are increasingly threatened by anthropogenic pressures, including climate change, overfishing, pollution, and ocean acidification (Cooley et al., 2023; Bellwood et al., 2004; Hughes et al., 2017). As these stressors intensify, coral cover and overall reef resilience decline, underscoring the need for effective conservation and restoration (Brandl et al., 2019; Bellwood et al., 2004). A critical component of reef conservation is habitat mapping and coverage analysis, which typically relies on expert taxonomists to annotate *in situ* underwater images (Hill & Wilkinson, 2004; Beijbom et al., 2015). However, this dependence on manual labeling severely limits the scale and speed of monitoring (Brandl et al., 2019; Bellwood et al., 2004). Machine learning (ML) offers a path to automation, but progress is impeded by two factors: (i) the scarcity of well-established, ML-ready datasets and benchmarks, and (ii) persistent domain shift, whereby classifiers degrade on sites with different photographic conditions, local assemblages, or label conventions (Belcher et al., 2023; Chen et al., 2021).

CoralNet (Beijbom et al., 2015; Chen et al., 2021), one of the leading annotation platforms, fine-tunes a global model on user-defined "sources." However, these source-specific classifiers frequently show poor performance on unseen sites owing to variations in taxonomy, imaging protocols, and local reef composition (Williams et al., 2019). ReefCloud.ai (AIMS, 2024) has improved label standardization but remains inaccessible to public researchers. To address these data and accessibility gaps, we introduce **ReefNet** as a complementary resource, providing a large-scale, publicly available coral reef imagery dataset with point-label annotations *taxonomically aligned* to the *World Register of Marine Species* (**WoRMS**) (Board, 2024). Figure 1 shows an overview of the dataset's structure. ReefNet unifies data from 76 carefully selected CoralNet sources and includes an additional 1.3K images

that we collected and annotated from the Al-Wajh lagoon in the Red Sea, comprising 4,609 expert annotations of Red Sea hard coral genera, offering a valuable contribution from an understudied biogeographic region. In total, the dataset contains $\sim$ 925K genus-level scleractinian coral annotations (Table 1). Rigorous manual verification by marine biologists was conducted on 8,962 annotations spanning various sources and labels. This process yielded a high-confidence repository of hard coral annotations across diverse biogeographic regions. In addition to point annotations, we enrich ReefNet with textual descriptions for each genus generated from scanned books (Veron, 2000a;b; Wallace, 1999), enabling language-grounded classification and supporting the development of vision-language models (VLMs).

Beyond dataset curation, **ReefNet** introduces two benchmarking configurations designed to reflect real-world deployment scenarios. **(i) Within-Source Benchmark (In-Distribution):** For each source, we split the dataset into train, validation, and test sets at the *image* level, ensuring no image appears in more than one split, even though individual images contain multiple sparse point annotations. This setup simulates how practitioners might train and evaluate models using locally labeled data from the same reef site. **(ii) Cross-Source Benchmark (Out-of-Distribution):** The model is trained on data from a subset of sources and evaluated on entirely distinct sources, explicitly addressing the *domain shift* challenge of applying models to unseen reef sites. Together, these complementary benchmarks reveal the substantial performance degradation under domain shift, as shown in our empirical results (section 4). Models evaluated in Table 3 consistently struggle with rare classes and morphologically similar taxa, underscoring persistent challenges related to class imbalance and fine-grained discrimination. By releasing the ReefNet benchmark, code, and pre-trained models, we aim to catalyze the development of more robust, domain-adaptive machine learning solutions for urgent reef monitoring applications.

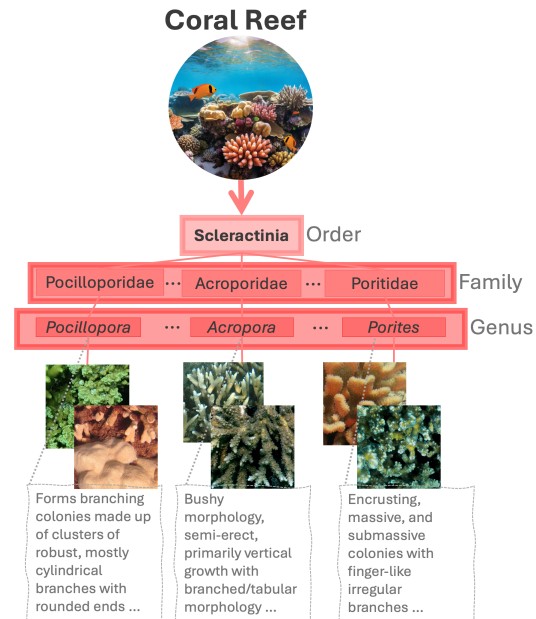

Figure 1: **Overview of the Hierarchical Structure of ReefNet**: a curated dataset of coral reef annotations focused on hard corals (*Scleractinia*), with textual descriptions for each genus.

ReefNet's global coverage, expert-curated labels, and flexible taxonomic structure provide a valuable foundation for accelerating innovation in coral-reef conservation. In summary, the contributions of this paper include: **(i) ReefNet**, a large-scale coral benchmark with WoRMS-aligned, standardized point-label annotations across 76 CoralNet sources. **(ii) Manual expert verification** and quality filtering for high-confidence benchmark splits. **(iii) Two benchmark settings**, *within-source* and *cross-source*, capturing in-domain and domain-shift classification performance. **(iv)** A new **Al-Wajh Lagoon dataset** covering an understudied Red Sea region. **(v)** Evaluation of **fine-tuned and zero-shot models** on both benchmarks.

## 2 RELATED WORK

Several large-scale machine-learning datasets have recently emerged to facilitate biodiversity classification. One example is *TreeOfLife-10M* (Stevens et al., 2023c), which consolidates imagery from iNat2021 (Horn et al., 2021), BIOSCAN-1M (Gharaee et al., 2024), and the Encyclopedia of Life (eol.org), surpassing 10 million images of terrestrial and marine organisms. Despite its scope, underwater taxa remain underrepresented in such general biodiversity repositories, largely because of the substantial logistical challenges associated with acquiring and annotating in situ marine imagery. To address this gap, *BenthicNet* (Lowe et al., 2024) was introduced, aggregating seafloor images from multiple surveys, including the Seaview Survey Photo-quadrat (González-Rivero et al., 2019) and Reef Life Survey (Edgar et al., 2020), among others (Friedman, 2020). BenthicNet constitutes a major step toward comprehensive marine coverage, but it primarily supports broad-scale benthic

Table 1: **Comparison With Existing Coral Classification Datasets.** *Image counts reflect availability as of May 13, 2025. Hard coral/genus-level annotations are not specified for CoralNet, since its labels are not standardized. †MosaicsUCSD includes 16 annotated orthomosaics, each from ~1.5K images. ‡CoralSCOP (Zheng et al., 2024) contains 330,144 binary coral/non-coral masks (not limited to hard corals). CoralVOS (Ziqiang et al., 2023) likewise provides only binary coral masks, so a hard coral count is N/A.

| Dataset | Geographic coverage | Annotation type | Lowest taxonomic level | Mapped to WoRMS | Number of images | Number of annotations | |
|---|---|---|---|---|---|---|---|
| | | | | | | Hard corals | Genus-level annotations |
| CoralNet (Beijbom et al., 2015) | **World** | Sparse points | **Species** | No | 4,524,792* | N/A | No |
| CoralSCOP (Zheng et al., 2024) | **World** | Masks | Order | No | 41,297 | 330,144‡ | No |
| CoralVOS (Ziqiang et al., 2023) | 17 sites (South China Sea) | **Masks** | Order | No | 60,456 | N/A | No |
| MosaicsUCSD (Edwards et al., 2017) | Palmyra | **Masks** | **Species** | No | 16† | 44,008 | **Yes** |
| Eilat (Raphael et al., 2020) | Eilat | Sparse points | Genus | No | 212 | ~12,000 | **Yes** |
| BenthicNet (Lowe et al., 2024) | **World** | Sparse points / image label | **Species** | Yes | **11,408,887** | 287,181 | **Yes** |
| Coralscapes (Sauder et al., 2025) | Red Sea (5 countries) | **Masks** | N/A | No | 2,075 | N/A | No |
| **ReefNet (ours)** | **World** | Sparse points | Genus | Yes | 181,223 | **924,626** | **Yes** |

habitat labeling using the CATAMI classification scheme (Althaus et al., 2015), and it also supports the WORMS (Board, 2024)taxonomy as it has 887533 annotations of which 287K are hard coral ('Scleractinia') annotations.(Table 1). Other coral-specific datasets have been proposed to support tasks such as dense segmentation. For instance, *CoralVOS* and *CoralSCOP* (Ziqiang et al., 2023; Zheng et al., 2024) offer pixel-level masks but generally categorize corals only at the order level (*Scleractinia*). Consequently, their utility for fine-grained ecological studies is limited. In contrast, *MosaicsUCSD* and *Eilat* (Edwards et al., 2017; Beijbom et al., 2016; Alonso et al., 2019) provide annotations at genus or species levels but are geographically constrained to a single reef site, reducing their ability to generalize across regions. Meanwhile, *CoralNet* (Beijbom et al., 2015) hosts a large volume of benthic imagery with both human- and machine-generated labels, but does not typically map them to an accurate, globally recognized taxonomy such as the WoRMS. *Coralscapes* (Sauder et al., 2025) extends benthic classification to a sizeable Red Sea dataset with over 170K polygon annotations, focusing on habitat cover types rather than genus- or species-level coral identification. **ReefNet** builds on prior efforts while addressing several key limitations, providing one of the most comprehensive collections of expert-verified coral annotations available. These annotations are mapped to the genus level across a diverse, global set of reef sites, addressing challenges like limited taxonomic resolution, geographic bias, and inconsistent labeling standards. By aligning its taxonomy with the World Register of Marine Species (WoRMS), ReefNet ensures consistency and compatibility with ongoing coral biodiversity research, supporting a broad spectrum of ecological and machine learning tasks from habitat classification to genus-level analysis. We provide further clarification of our advantages over CoralNet in appendix A.1.

## 3 DATA COLLECTION AND LABEL MAPPING

ReefNet is the result of a multi-stage, expert-guided curation process applied to publicly available benthic imagery and annotations from CoralNet. This section outlines the construction of the core dataset, followed by the label standardization pipeline and manual verification process that underpin its taxonomic reliability and ecological utility.

### 3.1 DATA COLLECTION METHODOLOGY

We started with **1,366** publicly available CoralNet sources and applied a series of semantic, ecological, and technical filters to identify high-quality reef datasets with taxonomically meaningful annotations. This involved removing test or incomplete sources, selecting only expert-verified labels, and retaining sources with sufficient image and annotation counts. We then applied domain-specific criteria, such as the presence of reef-building corals and shallow, *in situ* imagery, yielding **76** public sources with approximately ~**920K** genus-level coral annotations. The full source selection pipeline is detailed in appendix A.2.1.

**Al-Wajh Lagoon Data.** We additionally contribute a new dataset from the Al-Wajh Lagoon in the Red Sea (25.6°N, 36.8°E), comprising **1.3K** high-resolution *in situ* images and **4,609** expert annotations of hard corals. This dataset serves as a test set for the cross-source benchmark to evaluate fine-grained classification and cross-region generalization. Data collection details are provided in appendix A.2.6.

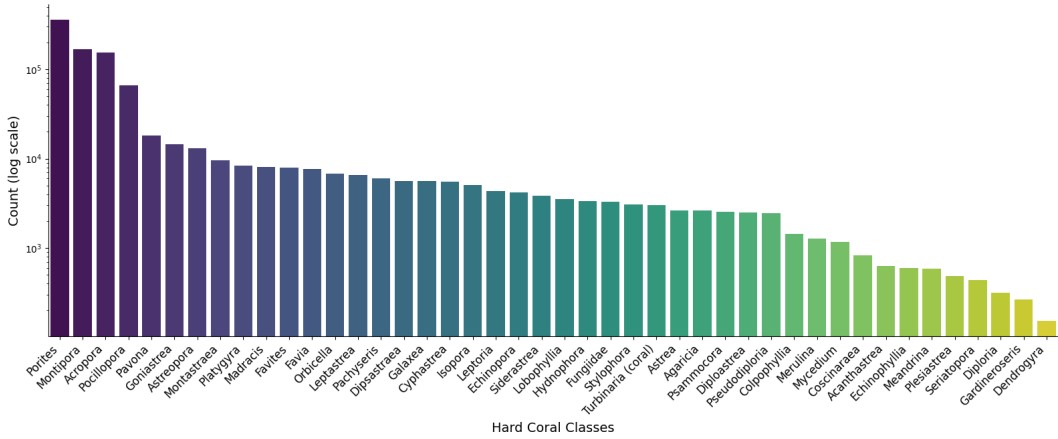

Figure 2: **Log-scale Distribution of Annotations Across Hard Coral Taxa.** The plot includes 43 genera and one family-level class (Fungiidae).

## 3.2 Label Standardization and Mapping

To ensure taxonomic traceability and biological consistency, we mapped annotation labels to the World Register of Marine Species (WoRMS Board (2024)). This step was applied, where applicable, to annotations from the 76 curated CoralNet sources and the Al-Wajh lagoon dataset. Hard coral labels were manually aligned with canonical scientific names and corresponding AphiaIDs[1]. This mapping enabled consistent aggregation of biological entities across sources and ensured compatibility with other biodiversity databases. The initial mapping revealed ∼920k annotations at multiple taxonomic levels, consolidated to genus-level hard corals. The dataset covers 44 reef-building hard coral (Order: *Scleractinia*) genera spanning 20 families. However, this initial label space exhibited inconsistencies in naming, taxonomic granularity, and semantic intent, necessitating further filtering and restructuring.

**Label Filtering and Semantic Consolidation.** To construct a benchmark dataset suitable for training generalizable AI models, we applied a biologically informed filtering strategy targeting syntactic variation (e.g., "Staghorn coral" vs. *Acropora cervicornis*) and taxonomic inconsistency (e.g., mixing species-, genus-, and family-level labels). This step was motivated by the inherent difficulty of fine-grained coral identification from imagery alone (Chen et al., 2021; Lowe et al., 2024), which often results in community datasets with inconsistent or imprecise taxonomic labeling.

Labels were retained only if they referred to hard corals at the genus level, except for *Fungiidae*, which was kept at the family level due to semantic similarity among its genera that makes genus-level verification difficult. Retained labels also had to meet the following criteria: appear at least 100 times in the dataset, be present in at least three distinct CoralNet sources with a minimum of 10 annotations per source, exhibit consistent and distinguishable visual patterns, and be taxonomically valid according to WoRMS. After filtering and consolidation, the ReefNet dataset used in our experiments includes 44 unique labels, grouped primarily at the genus level. This curated, taxonomy-aware structure enables hierarchical subsetting, scalable annotation, and custom label aggregation for ecological and machine learning tasks. The structured design of ReefNet is illustrated in Figure 1.

## 3.3 Dataset Statistics

### 3.3.1 Annotation Distribution

The ReefNet dataset comprises **334,162** images, of which **181,223** contain hard coral annotations, totaling **924,626** point annotations. These annotations span **26 marine ecoregions** across the tropical belt *sensu* (Spalding et al., 2007) (see appendix A.2.5 for details). The distribution is uneven: the Hawaiian Ecoregion alone contributes over 221K annotations, followed by the Samoa Islands (202K) and the Mariana Islands (108K). In contrast, regions such as the Southwestern Caribbean,

---

[1]An AphiaID is a unique identifier from the WoRMS database, linking each label to its corresponding entry in the hierarchical taxonomy. As coral taxonomy evolves, AphiaIDs allow users to verify the current validity of labels.

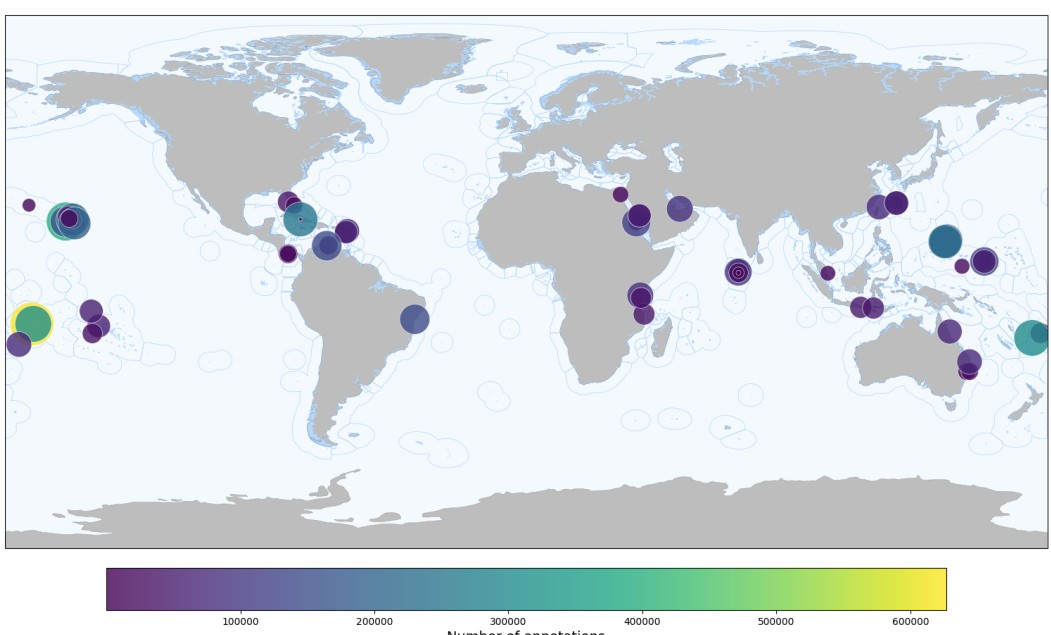

Figure 3: **Geographic Distribution of the Annotations.** Sources located within two degrees of Latitude and Longitude are grouped into a single point. One source did not contain any location data and is displayed in Antarctica. Marine Ecoregions of the World are shown in light blue (Spalding et al., 2007).

Floridian, and Eastern Brazil are sparsely represented, each with fewer than 8K samples. This geographic imbalance reflects broader disparities in coral monitoring efforts and may influence model generalization to underrepresented areas. Among all annotations, 921,351 are assigned to one of 44 classes, capturing a wide range of coral diversity. The most frequent genera, *Porites*, *Montipora*, *Acropora*, and *Pocillopora*, dominate shallow reefs globally (Figure 2, Figure 4), while several genera appear in less than 1K annotations, illustrating the dataset's imbalanced distribution.

### 3.3.2 COVARIATE DIVERSITY

In addition to the biological breadth of ReefNet, it exhibits substantial covariate diversity across environmental conditions, camera systems, and imaging protocols (Figure 3), leading to significant variations in resolution, lighting, and water quality across the data. For specific target deployments, it may be advantageous to only train on images from similar environmental conditions or acquisition parameters. We provide detailed metadata on geographic context and camera setups for each CoralNet source in appendix A.2.5, along with the metadata that will be publicly released, thereby enabling the exploration of optimal subsets.

### 3.4 HARD CORALS DATA QUALITY CONTROL

To assess annotation reliability, we conducted an expert review on a stratified subset of 8,962 hard coral samples. Each review involved evaluating images tied to specific *source–genus* pairs, with labels categorized as *Correct*, *Incorrect*, *Low Quality Image*, or *Hard to Decide*. A micro-averaged expert agreement percentage was calculated as the proportion of *Correct* labels among all reviewed samples. To support this process, we built a custom web-based application that enabled structured review of annotations across sources. Further details on the interface and the whole verification pipeline are provided in the supplementary material. Based on this expert feedback, showing an overall agreement rate of 73%, we performed targeted quality control, removing low-confidence genera, sources, or genus–source pairs. This affected a total of 920,017 hard coral annotations in the full dataset, guided by insights from the verified subset. These verification and filtering steps ensure that downstream models are trained on biologically reliable annotations and support ecologically valid interpretation of classification outcomes. Further details on the quality control are provided in appendix A.2.2 and appendix A.2.3.

**Source and Genus Filtering.** We first excluded any source where fewer than 50% of reviewed samples were deemed correct. Similarly, coral genus with less than 50% expert agreement across all verified sources were removed. This step retained 70 sources, with a post-filter expert agreement of 78%. After filtering, 860,463 annotations remained across 39 unique hard coral genera.

**Source–Genus Filtering.** We then applied a stricter filter at the source–genus level, retaining only pairs with at least 70% expert agreement in their verified samples for each source-genus pair. This resulted in a high-confidence subset with **92%** expert agreement on the representative verification set. After this step, **479,027** annotations were retained from 68 sources, while still preserving all 40 hard coral genera. A summary of the filtering process and benchmark splits is presented in Table 2.

## 4  REEFNET BENCHMARKS

A key practical challenge for coral-reef AI is the domain gap encountered when models trained on imagery from one dataset (a "source") are applied to imagery collected by another dataset. Models trained on datasets from specific locations frequently exhibit poor generalization to ecologically or geographically distinct areas (Williams et al., 2019; Belcher et al., 2023; Wyatt et al., 2025). Variations in camera setups, water clarity, depth, and annotation protocols, even for the same taxa, can degrade performance. To systematically evaluate model robustness and practical applicability in genus-level coral classification, we define two primary benchmarking scenarios: **(i) Within-source fine-tuning:** Images from a single source (Source A) are partitioned into distinct training, validation, and test subsets. Models are trained on a labeled subset and evaluated on the remainder. Although used by platforms such as CoralNet and ReefCloud, this method requires site-specific manual annotations, limiting scalability. **(ii) Cross-source deployment:** Models trained on all available images from Source A are directly evaluated on a different source (Source B), without further fine-tuning. This setting assesses generalization across domains, highlighting the model's ability to overcome domain shifts. ReefNet provides benchmarks for both settings with different training and testing configurations. A quantitative overview of all the splits is in Table 2.

### 4.1  REEFNET WITHIN-SOURCE SPLIT

For the within-source setting, we consider two splits that share an identical label set but with different expert agreement quality. In each split, each source is divided on the image level to prevent information leakage using Multilabel Stratified Shuffle Split (Sechidis et al., 2011; Brady, 2017). The two data-quality variants considered: **(i) Train-S1 / Test-S1**, with no source–genus filtration, of 81 % expert agreement, 802,956 training annotations from 70 sources; 9,972 validation annotations from 64 sources; 40,881 test annotations from 69 sources. **(ii) Train-S2 / Test-S2**, with source–genus filtration, of 92 % expert agreement percentage, 445,985 training annotations from 68 sources; 9,999 validation annotations from 63 sources; 40,881 test annotations from 66 sources.

### 4.2  REEFNET CROSS-SOURCE SPLIT

**Source Allocation.** Applying the quality control thresholds ($> 50\%$ expert agreement per source and per genus, and $> 70\%$ per source–genus pair) yields 68 eligible sources. We assign 44 to training (depending on the training variant), 17 to validation, and 7 to testing. Test sources are selected first to prioritize high expert agreement and maximize class overlap with training; validation sources are then chosen to maintain overlap with both training and test sets. The held-out test set **Test-S3&S4** comprises ∼34k images from 7 unseen high-quality sources spanning the Atlantic and Indo-Pacific regions (33 classes, 96% expert agreement). Additionally, we considered the Al-Wajh dataset as a test source **Test-W** to further investigate cross-source generalizability on 12 Red Sea genera.

**Training Variants.** For cross-source experiments, we contrast two training partitions to illustrate the quantity–quality trade-off: **(1)** *Train-S3* (no source–genus filtration): 733,391 annotations, 39 classes, 45 sources, 80% expert agreement. **(2)** *Train-S4* (with source–genus filtration): 406,552 annotations, 38 classes, 44 sources, 91% expert agreement.

Across all splits, the hard coral labels span 33–39 taxa, except in Test-W, where only 12 taxa are evaluated. Overlap statistics in Table 2 confirm that both the validation and test sets of S3&S4 remain taxonomically diverse despite their smaller size.

Table 2: **Summary of Dataset Splits Across Experimental Settings.** "Annotations" and "Classes" show the number of annotations and the corresponding number of unique labels. "Expert agreement percentage" refers to the correct rate of the verified annotations. "∩ Train" shows overlapping class counts with the training set. The same test sets are used for both cross-source training settings.

| In/Cross | Source-genus select. | Split | Annotations | Classes | # Sources | % expert agreement | ∩ Train |
|----------|---------------------|-------|-------------|---------|-----------|--------------------|---------|
| In | ✗ | Train-S1 | 802,956 | 39 | 70 | 81 | 39 |
| | | Val-S1 | 9,972 | 37 | 64 | 81 | 37 |
| | | Test-S1 | 40,881 | 39 | 69 | 81 | 39 |
| In | ✓ | Train-S2 | 445,985 | 39 | 68 | 92 | 39 |
| | | Val-S2 | 9,999 | 38 | 63 | 92 | 38 |
| | | Test-S2 | 23,043 | 39 | 66 | 92 | 39 |
| Cross | ✗ | Train-S3 | 733,391 | 39 | 45 | 80 | 39 |
| Cross | ✓ | Train-S4 | 406,552 | 38 | 44 | 91 | 38 |
| | | Val-S3&S4 | 37,473 | 33 | 17 | 92 | 33 |
| | | Test-S3&S4 | 34,040 | 33 | 7 | 96 | 33 |
| Cross | Al-Wajh Test Set | Test-W | 4,606 | 12 | 1 | 100 | 12 |

## 5 EXPERIMENTS

### 5.1 EXPERIMENTAL SETUP

**Evaluated Models.** We evaluate three types of models across all benchmarks: (i) Large Scale Pre-trained models fine-tuned on our dataset (Tan & Le, 2019; Chen et al., 2021; He et al., 2016; Liu et al., 2022; Dosovitskiy et al., 2021; Bao et al., 2022; Liu et al., 2021; Touvron et al., 2021), (ii) Vision-Language Models (VLMs) evaluated in a zero-shot setting (Ilharco et al., 2021; Radford et al., 2021; Stevens et al., 2023a; Zhai et al., 2023), and (iii) Multimodal Large Language Models (MLLMs) also evaluated in a zero-shot setting (Bai et al., 2025). Model performance is evaluated using Macro Recall, sometimes referred to as the balanced accuracy score (Pedregosa et al., 2011). Macro Recall is particularly suited for our task as the ReefNet data shows a high degree of class imbalance 2. Additionally, for top-performing models, per-class recall, precision, and F1 scores are provided in appendices( A.4.3- A.4.5). Additionally, we fine-tune BioCLIP on our hierarchical dataset, down to the Genus level (referred to as BioCLIP-FT), to investigate its performance on hard coral hierarchical classification, as it reported good performance on other datasets spanning plants, animals, and fungi on TreeOfLife-10M (Stevens et al., 2023c). The details about the training setup for all models have been included in the supplementary material.

### 5.2 FINETUNING RESULTS

All models in Table 3 show substantial performance drops under cross-source evaluations, revealing challenges in generalizing across sources with varying imaging conditions. Persistent issues include poor recall on rare classes and confusion between morphologically similar genera, highlighting the need for models that better handle domain shifts, class imbalance, and fine-grained visual distinctions. While BioCLIP-FT achieves overall good performance in within-source evaluations, particularly on the higher-quality split, and ViT (MAE-pretrained) shows the strongest generalization in cross-source settings, the broader trend across Table 3 reveals a consistent struggle for all models to maintain accuracy under domain shift, underscoring a persistent challenge in generalizing coral classification models beyond their training distribution. This degradation is largely driven by the high variability in imaging conditions across different reef survey collections. In addition, performance remains especially poor on rare classes and visually similar genera. Many coral genera share subtle morphological traits, such as meandroid or polygonal corallite structures, which are difficult to distinguish in non-close-up survey images. Tackling these harder examples, as opposed to the easier, more distinct ones, will require further development in both model design and data curation strategies.

**Within-Source Splits:** The two within-source benchmarks in Table 3 demonstrate higher Macro Recall on the higher-quality split (Train-S2 / Test-S2) for all models except ViT (MAE-pretrained). This trend does not necessarily suggest improved performance due to enhanced data quality, as direct comparisons between Train-S1 and Train-S2 remain inconclusive due to the use of distinct test sets (Test-S1 and Test-S2). Nevertheless, the overall relative performance ranking of models remained consistent across both benchmarks. The stricter quality control applied to Test-S2 samples establishes

Table 3: **Hard Coral Classification Macro Recall Under Different Train/Test Settings.** Best scores are in **bold**; second best are underlined.

| Model | Train/Test split | | | | | |
| | within-source Train-S1 / Test-S1 | within-source Train-S2 / Test-S2 | cross-source Train-S3 / Test-S3 & S4 | cross-source Train-S4 / Test-S3 & S4 | cross-source Train-S3 / Test-W | cross-source Train-S4 / Test-W |
| --- | --- | --- | --- | --- | --- | --- |
| EfficientNet (Tan & Le, 2019) | 78.36 | 82.41 | 48.01 | 41.78 | 69.29 | 64.59 |
| ResNet (He et al., 2016) | 63.91 | 64.49 | 37.96 | 28.49 | 64.14 | 51.22 |
| ConvNext (Liu et al., 2022) | 77.52 | 82.88 | 42.21 | 47.06 | 69.19 | **68.08** |
| BEiT (Bao et al., 2022) | 72.18 | 73.63 | 45.56 | 32.17 | 64.24 | 62.31 |
| DeiT (Touvron et al., 2021) | 74.36 | 77.40 | 45.46 | 40.94 | 63.19 | 66.67 |
| Swin (Liu et al., 2021) | 77.71 | 82.64 | 45.37 | 42.33 | 65.64 | 65.41 |
| ViT-B (scratch) (Dosovitskiy et al., 2021) | 75.23 | 81.86 | 39.62 | 45.08 | 66.18 | 66.12 |
| ViT-B (Dosovitskiy et al., 2021) | 75.47 | 79.00 | 50.49 | 44.83 | 69.71 | 66.18 |
| ViT-B (MAE (He et al., 2022) pretrained) | **79.97** | 77.72 | **56.21** | **47.07** | **71.77** | 61.40 |
| BioCLIP-FT (Stevens et al., 2023b) | 79.32 | **84.06** | 45.31 | 42.94 | 70.07 | 63.42 |
| Number of test classes | 39 | 39 | 33 | 33 | 12 | 12 |
| Number of test samples | 40,881 | 23,043 | 34,040 | 34,040 | 4,606 | 4,606 |

the (Train-S2 / Test-S2) split as a more reliable benchmark compared to (Train-S1 / Test-S1), as higher-quality test data yields more statistically meaningful measures of performance. In these within-source experiments, we found that BioCLIP-FT notably outperformed other models, performing best on the (Train-S2 / Test-S2) split with 84.06%. This result highlights the potential benefits of leveraging pretrained knowledge from large-scale taxonomic biological data in TreeOfLife-10M (Stevens et al., 2023c).

**Cross-Source Splits:** To investigate the trade-off between training data quantity and quality, the two cross-source benchmarks in Table 3 share a common test set (Test-S3&S4). Train-S3 contains approximately 0.7 million training samples with an 80% expert agreement percentage, while Train-S4 comprises around 0.4 million samples with a 91% expert agreement percentage. Unlike the within-source benchmarks, the two cross-source benchmarks do not exhibit consistent relative performance rankings across models, nor do they reflect an overall ranking similarity with the within-source splits. Nonetheless, macro recall scores for all models show substantial degradation in cross-source evaluations, with drops up to 41% for BioCLIP-FT compared to the within-source benchmark (Train-S2/Test-S2). This performance decline is expected, as the cross-source setup introduces domain generalization challenges due to high variability across different reef surveys.

Notably, BioCLIP-FT fails to maintain its top performance in this setting, highlighting the difficulty of cross-source generalization even when the model is aligned with the taxonomic structure of the dataset. Interestingly, ViT (MAE-pretrained) demonstrates stronger generalization capabilities, achieving 56.21% macro recall on Test-S3&S4. This result demonstrates the advantage of self-supervised pretraining on large datasets. Additionally, comparing the performance of ViT (MAE-pretrained) trained on Train-S3 versus Train-S4 reveals the benefit of increased training data quantity, even when the data quality is lower (80% vs. 91% expert agreement percentage). Testing on the Al-Wajh source (Test-W, with 12 taxa) also showed improved performance with more training data. Its higher scores, compared to other cross-source results, are due to most test classes being abundant in the training set.

**Loss Function and Class Imbalance.** Rare classes exhibit lower recall, an expected outcome given the class imbalance in the training data (see per-class analysis in appendix A.4.3). To mitigate this, we evaluated alternative loss functions. As presented in Table 4, training ViT-L with class-balanced cross-entropy (Cui et al., 2019) did not yield improvements in any split. In contrast, focal loss (Lin et al., 2017) consistently outperformed standard cross-entropy, with further gains observed when using its class-balanced variant (Cui et al., 2019).

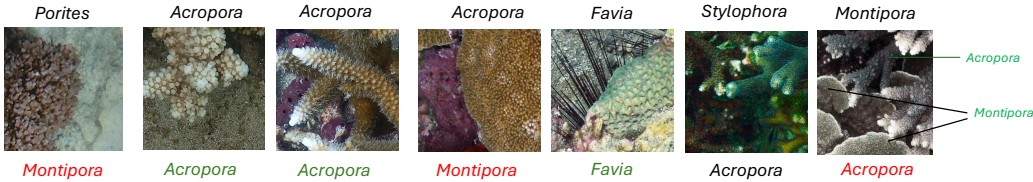

Figure 4: **Qualitative examples.** Ground truth is shown above and model prediction below; the rightmost image illustrates the challenge of assigning a single label to patches with multiple corals.

Table 4: **Loss Function Ablations Using ViT-L-384 (macro recall).**

| Loss | within-source Train-S1 / Test-S1 | within-source Train-S2 / Test-S2 | cross-source Train-S3 / Test-S3&S4 | cross-source Train-S4 / Test-S3&S4 |
|---|---|---|---|---|
| CE | 77.37 | 80.71 | 50.14 | 44.98 |
| CB-CE (Cui et al., 2019) | 76.58 | 76.23 | 46.96 | 42.32 |
| Focal (Lin et al., 2017) | 78.39 | 84.33 | 46.33 | 41.24 |
| CB-Focal (Cui et al., 2019) | **81.86** | **85.71** | **52.87** | **45.65** |

## 5.3 ZERO-SHOT RESULTS

From Table 5, it is evident that all models struggle with the zero-shot classification task, particularly when compared to the fine-tuned models discussed in section 5.2. This performance gap is expected, as most of the zero-shot models have likely encountered limited or no coral-specific data during pretraining. However, BioCLIP (Stevens et al., 2023a), which achieves the best zero-shot performance, has been pre-trained on the TreeOfLife-10M (Stevens et al., 2023c) dataset, which includes a wide range of biological species, including corals. This pretraining provides BioCLIP with prior knowledge of coral-related visual features. Therefore, the performance of VLMs is consistent with their training data exposure.

We also evaluate Qwen2.5-VL (Bai et al., 2025) under different configurations. Directly prompting Qwen2.5-VL to identify the coral genus from an image results in subpar performance. To improve this, we provided additional context by generating coral genus descriptions using GPT-4.0; we refer to this setup as Qwen-GPT. While this improved Qwen2.5-VL's performance on the in-source split, the performance on the cross-source split remained unchanged. Upon inspection, we found that the GPT-generated descriptions contained repetitive and generic content, limiting their utility. To address this issue, we curated high-quality textual sources by extracting coral-specific information from three reference books (Veron, 2000a;b; Wallace, 1999). These excerpts were then summarized by GPT-4.0 into discriminative genus descriptions, which were provided as context to Qwen2.5-VL; we refer to this setup as Qwen-Book. This approach improved Qwen2.5-VL's performance on both in-source and cross-source splits. Although Qwen2.5-VL still lags behind BioCLIP, these results highlight the potential of using high-quality, domain-specific textual context to enhance coral recognition in MLLMs. We believe that further refining the descriptions and retrieval strategies could significantly close the gap.

Table 5: **Zero-Shot Classification.** Macro recall of open-source VLMs on hard corals classification. Qwen-GPT uses GPT-4o-generated genus descriptions; Qwen-Book adds GPT-4o summaries of domain-specific books (Veron, 2000a;b; Wallace, 1999).

| | CLIP | SigLIP | OpenCLIP | BioCLIP | Qwen2.5-VL | Qwen-GPT (GPT-4o genus desc.) | Qwen-Book (GPT-4o book summ.) |
|---|---|---|---|---|---|---|---|
| Within-source Test-S2 | 2.50 | 3.31 | 3.94 | **9.92** | 2.98 | 4.15 | 6.04 |
| Cross-source Test-S3&S4 | 1.00 | 1.91 | 4.56 | **10.33** | 3.92 | 3.92 | 6.27 |

## 6 CONCLUSION

We introduced ReefNet, a globally curated benchmark comprising 925K genus-level hard coral annotations mapped to the WoRMS taxonomy. Spanning diverse biogeographic regions and featuring a standardized, expert-reviewed label set, ReefNet serves as a foundational resource for automated coral monitoring. Our benchmarks show that models trained on within-source data achieve strong performance, but significant drops under domain shift highlight challenges in generalizing to unseen reef systems, especially for rare taxa. Notably, some common coral genera were detected with over 90% accuracy across distinct test locations, demonstrating the potential for generalization to widespread classes. ReefNet-trained models can also be leveraged to pre-label common genera, streamlining expert annotation and enabling efficient local model adaptation. By releasing the ReefNet benchmark, code, and models, we aim to support the development of scalable, domain-adaptive tools for coral reef monitoring and conservation. Further details on the dataset, the verification process, and additional experiments are provided in the supplementary material.

## 7 ETHICS

The primary goal of this research is to advance AI for a positive societal impact, specifically in the domain of biodiversity conservation and reef ecology. Our work introduces a new benchmark, ReefNet, which is constructed from publicly available images and textual data sourced from encyclopedic resources. All data used will be made publicly available, adhering to the licensing terms of its original source. The dataset contains images of animal species, corals, and other marine life and does not involve human subjects, thus presenting no personal data privacy concerns.

We acknowledge that all large-scale datasets are susceptible to inherent biases. Our benchmark may reflect geographic and taxonomic biases present in the publicly available data it is derived from. Similarly, the language models used for generating and distilling descriptions (e.g., GPT-4o) may carry their own latent biases. We have sought to mitigate this by involving reef ecologists in our data curation process. We believe the potential for misuse of this technology is low, as its primary application is intended for scientific research and environmental monitoring.

## 8 REPRODUCIBILITY

We ensure reproducibility of our work. Our dataset will be made publicly available under appropriate licenses. For evaluation, we specify hyperparameters, architectures, and training details in the main text and supplementary material. Our codebase, including data loaders, evaluation scripts, and fine-tuning implementations, will be released on GitHub. Random seeds are fixed in all experiments, and we report results across multiple runs where applicable. Together, these steps ensure that our results can be independently verified and extended by the community.

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

# A  APPENDIX

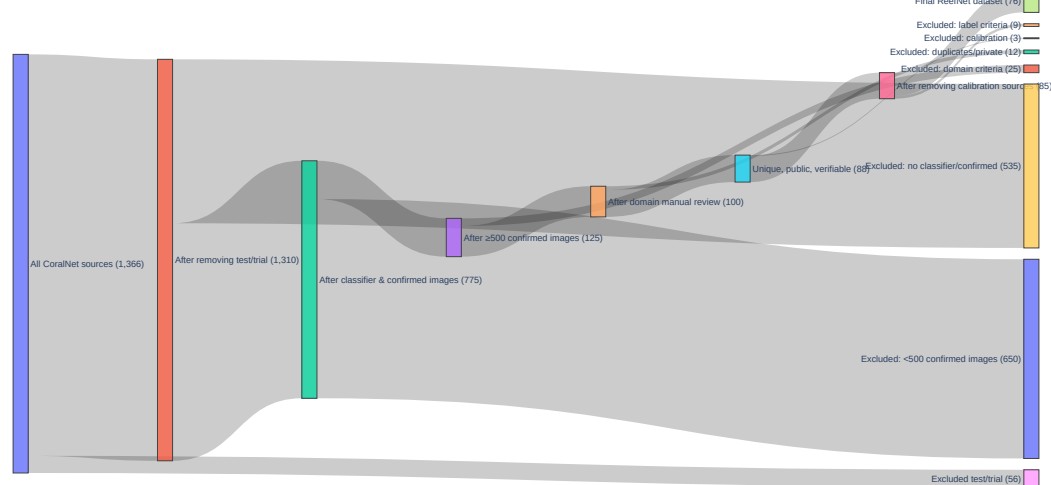

Figure 5: **ReefNet Curation Pipeline.** The diagram illustrates the sequential filtering and exclusion steps applied to 1,366 CoralNet image sources to arrive at the final ReefNet dataset of 76 sources. Sources were excluded due to insufficient number of human-verified annotations, ecological or privacy concerns, and calibration-related issues.

## A.1  REEFNET IN RELATION TO CORALNET

Despite CoralNet's (Beijbom et al., 2015) wide adoption as a platform for coral annotations and publishing datasets, it presents several challenges that limit its usability for ML practitioners to train and evaluate ML models in a straightforward way, unlike other biological fields where the community has built well-established ML-ready datasets(e.g., CUB-200-2011, NABirds, iNaturalist, IP102) (Wah et al., 2011; Van Horn et al., 2015; 2018; Wu et al., 2019). ReefNet addresses these shortcomings, making it a more reliable resource for scientific exploration and algorithmic development. The key limitations of CoralNet are:

- **Lack of a unified label set across sources.** CoralNet aggregates data with heterogeneous and often incompatible label sets, making large-scale integration infeasible.

- **Absence of taxonomically verified hard coral labels.** Many sources use outdated, ambiguous, or generic labels instead of scientifically recognized names (e.g., those in the World Register of Marine Species).

- **No standardized quality control.** Without systematic quality checks, it is difficult to identify reliable samples, which is critical for both training and evaluation.

- **No large-scale benchmark framework.** CoralNet does not provide a standardized evaluation setting for consistent model comparison.

To address these challenges, we propose ReefNet with the following key contributions:

- **World Register of Marine Species (WoRMS) (Board, 2024)-based unified taxonomic labeling:** ReefNet adopts a unified labeling scheme grounded in WoRMS to integrate heterogeneous sources. This harmonization also enabled the inclusion of a Red Sea–focused dataset. We anticipate that this taxonomy will encourage future work to adopt scientifically sound annotation practices.

- **Rigorous re-verification:** CoralNet sources were annotated by distinct groups with variable expertise, leading to inconsistencies in label provenance. While acceptable within individual sources, these inconsistencies hinder multi-source integration. ReefNet mitigates this issue by introducing a centralized verification step: a dedicated review team, composed of individuals with complementary

backgrounds, re-evaluated all included sources. This ensures a consistent, cross-source quality standard unattainable through direct dataset merging.

**- Standardized benchmarks:** ReefNet establishes reproducible benchmarks with clearly defined splits, including: i) *Within-source split:* Evaluates performance when training and testing distributions are closely aligned, ii) *Cross-source split:* Assesses robustness to domain shifts arising from differences in imaging tools, weather, depth, and other capture conditions, iii) *Expert agreement splits:* Enable controlled comparisons between training with larger, noisier datasets and smaller, cleaner ones, highlighting the trade-off between data quantity and label quality.

**- Red Sea–focused benchmark:** A dedicated Al-Wajh split allows systematic evaluation of model generalization on an ecologically critical but underrepresented region, providing a benchmark for studying regional biases and adaptation.

## A.2 ADDITIONAL INFORMATION ON THE DATASET

### A.2.1 REEFNET DATA COLLECTION METHODOLOGY

The **ReefNet** dataset was constructed through a multi-stage curation pipeline applied to publicly available sources hosted on *CoralNet*. Beginning with all **1,366 publicly listed CoralNet sources**, we applied a series of semantic, ecological, and technical filters to isolate high-quality, taxonomically relevant reef imagery and annotations. These steps ensured that retained sources featured dense, consistent, and biologically meaningful annotation data.

We first excluded sources with names containing keywords such as *"test"* or *"trial"*—commonly created by users trialing the platform—reducing the pool to **1,310 sources**. An exception was made for *CoralNet Assistance Test*, which was retained after manual inspection confirmed its annotation reliability and ecological relevance.

Next, we removed sources lacking a trained classifier or any *confirmed* (i.e., human-annotated) images, yielding **775 sources**. Applying a minimum threshold of **500 confirmed images** further reduced the set to **125 sources**. These were manually reviewed against two domain-specific criteria: (i) the presence of annotations for **Scleractinian** (reef-building hard coral) taxa, and (ii) use of **in situ** imagery from shallow, tropical reef environments, resulting in **100 sources**.

To ensure uniqueness, we eliminated duplicate annotations based on filename, label identity, and point-level annotation coordinates—occasionally leading to the exclusion of entire sources. We also removed any sources that had become private after our data crawl, yielding **88 public, verifiable sources**. Following consultation with data owners, we excluded three calibration sources used for training novice annotators, resulting in **85 sources**.

A final filtering stage retained only labels that met all of the following criteria: (i) ecologically relevant (i.e., hard coral taxa), (ii) appeared at least 100 times, (iii) were present in a minimum of three distinct sources with at least 10 annotations per source, (iv) exhibited consistent, visually distinguishable patterns, and (v) were taxonomically valid per WoRMS. After filtering and consolidation, the final **ReefNet** dataset consisted of **76 sources** and approximately **925,000 hard coral annotations**.

In parallel, we standardized associated metadata for each source, including geographic location, contributor affiliation, and source history (see Table 13). The complete metadata will be made available as a CSV file.

### A.2.2 REEFNET DATA QUALITY CONTROL

To ensure taxonomic reliability and consistency across ReefNet annotations, we implemented a multi-stage expert verification and filtering process. This process was supported by a custom web-based application developed specifically for large-scale, structured review of coral genus annotations by marine biologists. Expert feedback collected through this platform was used to assess annotation quality, identify systematic labeling errors, and curate high-confidence subsets for benchmark construction.

**Manual Verification Tool.** We developed a custom web-based platform to support expert review of coral genus annotations. Marine biologists used the tool to verify stratified subsets of model predictions across sources and taxa. Each annotation was labeled as *Correct*, *Incorrect*, *Low Image*

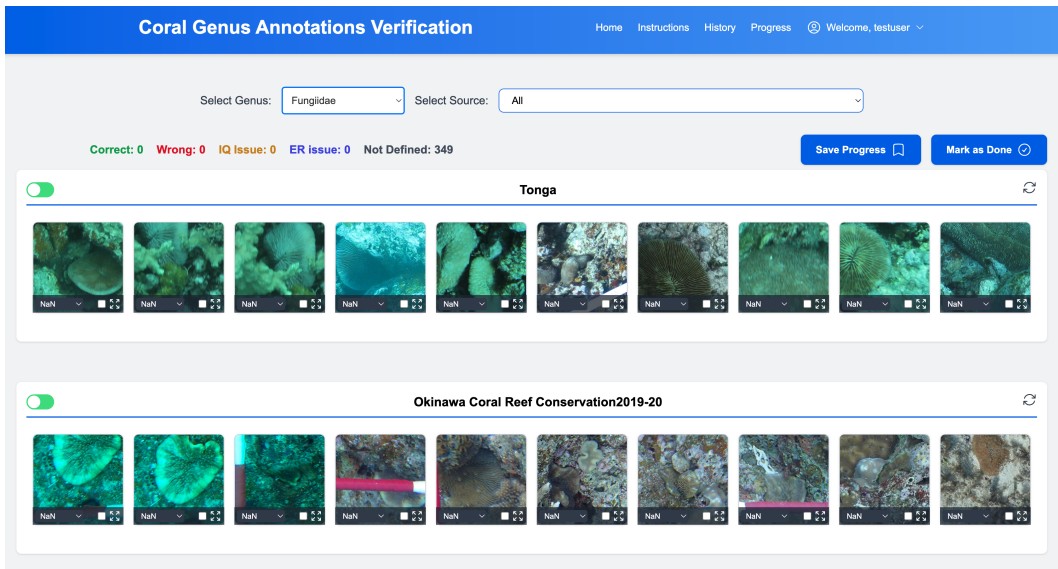

Figure 6: **Screenshot of the ReefNet manual verification platform.** Experts assess genus-level annotations by reviewing image patches with full-context overlays and assign structured labels via a dropdown interface. The tool supports efficient, large-scale quality control of AI-generated coral labels.

*Quality*, or *Hard to Decide*, based on a zoomable patch and full-image context. The interface supports genus- and source-specific filtering, progress tracking, and pseudonymized user sessions. The system is lightweight, scalable, and integrated with the ReefNet pipeline. All verification actions are logged with metadata for auditability and downstream filtering. Expert feedback informed exclusion criteria and helped construct benchmark splits (e.g., Train-S2, Train-S4) with up to 92% agreement. These labels were also used to refine taxon definitions and guide model re-training.

### A.2.3    MORE DETAILS ON THE CENTRALIZED RE-VERIFICATION PROTOCOL

Our reviewers are not inherently superior to the original CoralNet annotators. However, our audit revealed systematic issues such as taxonomic inconsistencies across sources, frequent mislabeling (e.g., genus-level errors in *Acropora*), and the absence of standardized label conventions. Moreover, because each CoralNet source was annotated by a distinct group with varying taxonomic expertise, intra-source labels are acceptable but introduce significant noise and bias when merged across datasets. ReefNet addresses this by applying centralized expert verification across all sources, thereby establishing a consistent cross-source quality baseline that CoralNet alone cannot provide.

**Expert Reviewers Background.** The verification team included one highly experienced coral taxonomist, three senior coral ecologists, and six PhD-level specialists in coral systematics and reef monitoring. Full reviewer details will be provided in the acknowledgments after the review process to preserve anonymity.

**Review Protocol.** We implemented a stratified random sampling procedure covering 8,962 patches (10 per genus per source). Each reviewer independently annotated the selected patches and flagged uncertain samples. Unresolved cases were either excluded or explicitly marked as low-confidence. This protocol increased expert agreement from 73% to 92% by retaining only source–class pairs that received high-confidence consensus across verifiers.

**Goals of Expert Filtering.** The objective of our verification is not to override CoralNet annotations, but to augment them with standardized, reproducible curation. Specifically, the process (i) standardizes labels through WoRMS AphiaIDs, (ii) resolves cross-source inconsistencies, and (iii) produces a high-confidence dataset suitable for benchmarking machine learning models in ecological research.

**Scalability Considerations.** Although the verification tool enables efficient, structured feedback from expert reviewers, its scalability remains bounded by the availability of qualified taxonomists.

As ReefNet grows or is applied to more diverse regions, expanding human-in-the-loop verification to support broader annotation throughput will be necessary. Future directions include active learning to prioritize ambiguous or low-confidence cases, hierarchical review pipelines with tiered expertise, and exploration of consensus-based crowdsourcing for preliminary filtering stages. Addressing these limitations will be critical to sustaining high-quality annotation pipelines in large-scale ecological monitoring.

### A.2.4 CORALNET POINT SAMPLING STRATEGIES

Randomized point sampling is a long-established ecological standard and has also been widely adopted in terrestrial ecology (e.g., canopy cover, bird populations, vegetation structure). While ReefNet models treat each annotation point as an independent patch-level sample, the underlying point annotations originate from CoralNet-selected sources, which employ automated sampling methods across sites with:

- **Simple random:** 51 sources
- **Stratified random:** 23 sources
- **Uniform grid:** 2 sources

ReefNet includes a total of 925,322 point annotations across 181,046 images, with the following per-image statistics:

- **Mean:** 40 points per image
- **Median:** 22 points per image
- **Range:** 25–180 points per image

### A.2.5 ADDITIONAL DETAILS ON COVARIATE DIVERSITY

This section highlights three key aspects of metadata in ReefNet that are essential for both machine learning and biological applications: i) geographic location (Figure. 3), ii) image white balance, and iii) image resolution.

**Ecoregions Distribution.** To assess the geographic and ecological diversity of ReefNet, we grouped all annotations by marine ecoregion using the Marine Ecoregions of the World (MEOW) classification system. Figure 7 summarizes the distribution of sources, images, and annotations across 25 unique ecoregions.

The dataset spans a wide range of coral reef habitats, including the Red Sea, Caribbean, Central Indo-Pacific, and Central Pacific. Notably, several biodiversity hotspots are heavily represented: the Hawaiian Islands (18 sources, 221,419 annotations), Samoa Islands (2 sources, 201,685 annotations), and Mariana Islands (6 sources, 107,553 annotations). Other regions, such as the East Caroline Islands and Fiji, also contribute substantial volumes of data, enriching the taxonomic and ecological breadth of the dataset.

The number of image sources per ecoregion varies significantly, reflecting uneven global monitoring efforts. While some ecoregions are represented by multiple contributors and thousands of images, others—such as the Tweed-Moreton or Southwestern Caribbean—appear undersampled. This variation is important for evaluating model robustness and generalization across biogeographically distinct environments.

All counts are shown on a log scale to accommodate differences spanning several orders of magnitude. Overall, this distribution highlights ReefNet's potential to support geographically robust coral classification and cross-region generalization studies.

**Image White Balance.** Varying water clarity, depth, and light penetration in combination with the use of different camera configurations or settings can lead to strong color casts in underwater imagery. Therefore, we document the average intensities of the Red, Green, and Blue color bands across CoralNet sources (Figure 15). Sources with non-overlapping peaks in their RGB distributions generally suffer from color casts due to poor white balancing. The variety of color casts captured in the ReefNet dataset improves the robustness of models trained on ReefNet, however, certain

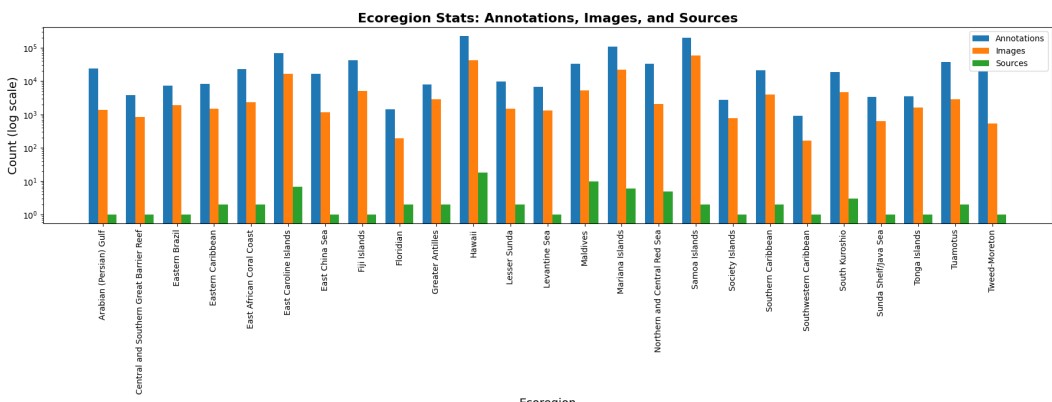

Figure 7: **Distribution of annotations, images, and sources across marine ecoregions.** Each bar group represents one of 25 marine ecoregions covered in the ReefNet dataset, as defined by the MEOW classification. The plot shows the number of annotations, distinct images, and image sources per ecoregion (log scale). Geographic coverage is notably high in regions like Hawaii, Samoa, and the Mariana Islands, but more limited in some Atlantic and Western Indian Ocean regions.

applications may benefit from using imagery with lighting conditions that closely match the target dataset.

**Image Resolution.** Image resolution in the ReefNet dataset ranges from 0.2 to 27 megapixels, with a median resolution of 12 megapixels. This diversity of image resolutions enhances the generalizability of AI models trained on ReefNet data. However, similarly to the white balance, users may prefer to extract images with a consistent resolution for specific tasks. Most sources in ReefNet exhibit uniform resolution distributions as a result of the use of standard camera systems. However, a few sources show a lot of variation, which generally stems from the manual cropping of the benthic quadrats visible in the image (Figure 16).

To facilitate the creation of customized sub-datasets, source-level metadata—including geographic location, RGB color profiles, and resolution statistics—will be made available with the ReefNet dataset on HuggingFace.

### A.2.6    AL-WAJH LAGOON DATASET

**Dataset Overview.** As part of a larger environmental monitoring effort, nearly 300 coral reefs were surveyed in the Al-Wajh lagoon (25.6°N, 36.8°E) between March and September 2021. Surveyed reefs were strategically selected using a stratified random design to ensure representation of the region's diverse reef habitats, including Reef Walls, Reef Crests, Reef Slopes, Patch Reefs, and Algal Reefs. The Al-Wajh Test dataset is composed of a subset of this imagery collected from the outer reefs of the lagoon. Imagery in the Al-Wajh Test dataset was collected using purpose-built photoquadrats, ensuring images were taken from a consistent distance from the seafloor. The camera was Canon PowerShot G9 X Mark II which shoots images of 20.2 MP. This represents the upper end of the resolutions available in the ReefNet dataset (Figure 16). A total of **1,376 images** were manually annotated on the CoralNet platform. Per image, 10 points were generated following a stratified random design of 2 rows and five columns, generating a total of **4,609** annotations of hard corals.

**Annotation Distribution.** The Al-Wajh Lagoon dataset covers 12 coral genera that overlap with those used to train the different models. Figure 9 shows the annotation distribution, which exhibits a bias toward more abundant classes such as *Porites*, *Acropora*, *Pocillopora*, and *Montipora*.

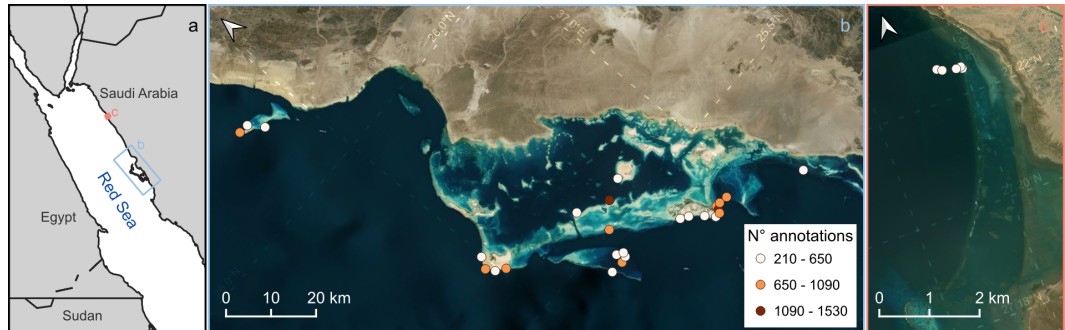

Figure 8: **Map showing the origin of the imagery in the Al-Wajh test dataset.** Panel a shows the location of panel b and c in the Red Sea. Panel b shows the Al-Wajh lagoon with the specific locations of the annotations in the test set. Panel c shows the location of a limited set of images taken outside the Al-Wajh lagoon. Map contains Natural Earth data and Bing satellite imagery.

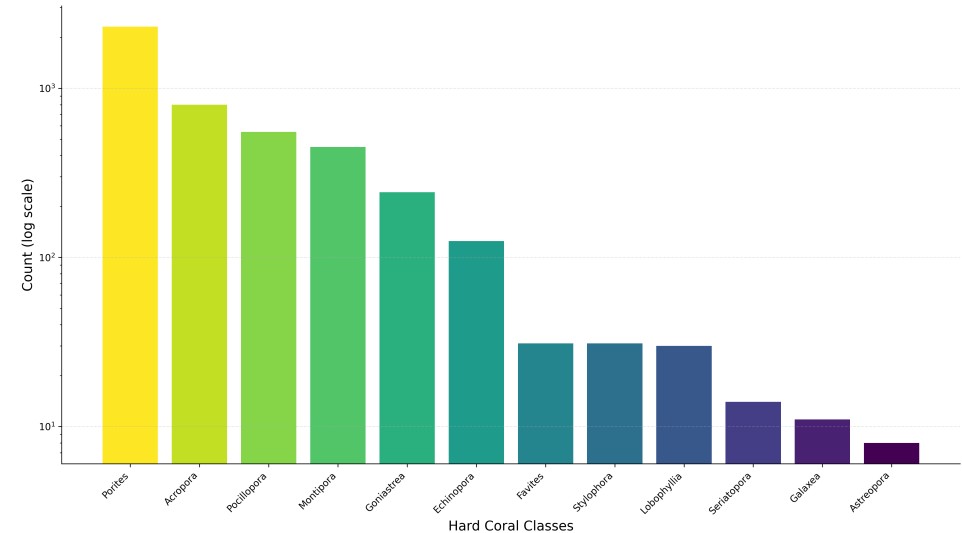

Figure 9: **Log-scale Distribution of Annotations Across 12 Hard Coral Genera in the Al-Wajh Lagoon Dataset.** The distribution shows a bias toward some classes, comparable to the distribution in ReefNet.

### A.3 EXPERIMENTAL DETAILS

#### A.3.1 TRAINING SETUP

**General Settings.** Unless otherwise specified, all models were trained for 100 epochs using the AdamW optimizer with a base learning rate of $2 \times 10^{-5}$ and a batch size adjusted per model according to memory constraints. Inputs were normalized using the ReefNet training set statistics: mean $[0.385, 0.419, 0.341]$ and standard deviation $[0.162, 0.177, 0.158]$. We applied standard image augmentation techniques including random horizontal flipping, color jitter, RandAugment (`rand-m7-mstd0.5-inc1`), Mixup (0.4), CutMix (0.5), and random erasing with a probability of 0.1 (`reprob`=0.1). Training was conducted with automatic mixed precision (AMP), channels-last memory format, and cosine learning rate scheduling with optional warmup. A held-out validation set was used to monitor performance during training and to select the best-performing model checkpoint across all the splits. No validation samples were included in training or augmentation procedures, ensuring a clean separation between training and evaluation.

**Architectures.** We evaluated eight supervised models: ViT-Large (pretrained and from scratch), DeiT-Base, Swin-Base, ResNet-101, ConvNeXt-Large, EfficientNet-B4, BEiT-Base, and BioCLIP.

Most models used pretrained weights from ImageNet-1K or their respective public checkpoints unless stated otherwise. Input resolutions varied by architecture: 224×224 for ViT/DeiT, 384×384 for ConvNeXt/BEiT, and 380×380 for EfficientNet. Model-specific batch sizes were selected to balance memory efficiency and throughput.

**Self-Supervised Models.** We evaluated a self-supervised variant using a Masked Autoencoder (MAE)He et al. (2022) with a ViT-LargeDosovitskiy et al. (2021) backbone. Pretraining was performed on 1.2 million $512 \times 512$ patches of hard coral annotations, using data from an earlier stage of the ReefNet pipeline prior to the final filtration step (i.e., using 85 sources, before removing nine low-quality sources). The MAE was trained for 800 epochs with a 75% masking ratio and cosine learning rate scheduling (base LR $1.5 \times 10^{-4}$).

Fine-tuning was conducted on the fully filtered ReefNet dataset over 100 epochs using a learning rate of $5 \times 10^{-4}$ and a layer-wise decay of 0.65. The model was trained using the same setup as supervised runs and evaluated across all four benchmark splits described in the main paper.

**BioClip Training Setup.** We fine-tuned BioCLIP using ReefNet's hierarchical taxonomic structure, initializing with pretrained weights from TreeOfLife-10M. For each genus, we constructed a taxonomy string by concatenating all hierarchical levels from kingdom to genus based on WoRMS (e.g., `"Animalia;Cnidaria;Hexacorallia;Scleractinia;Acroporidae;Acropora"`). BioCLIP was trained for 100 epochs using AdamW (learning rate $1 \times 10^{-4}$, weight decay 0.2) with cosine learning rate scheduling. We selected the best-performing checkpoint based on validation accuracy. The model was trained using local and global loss objectives, as in the original BioCLIP setup.

**Dataset Input.** All models were trained using $512 \times 512$ patches extracted around point annotations from full-resolution CoralNet imagery, referenced via a custom CSV loader. We selected this patch size to balance local texture and global morphological structure, both of which are essential for accurate coral genus classification. Preliminary experiments with smaller patches ($224 \times 224$ and $448 \times 448$) showed diminished performance, likely due to loss of spatial context or insufficient detail. This design choice aligns with prior work showing that effective coral classification requires capturing both textural and structural cues Almazán et al. (2019). Each training sample in our data includes genus annotation or family in the case of Fungiidae, patch coordinates, and source metadata.

**Loss Function Parameters** In our loss function ablation (Table 4), we evaluated several class-balanced and standard variants. We used **Cross-Entropy (CE) Loss** implemented with `PyTorch nn.CrossEntropyLoss()`. For **Focal Loss**, we set $\gamma = 2$ and $\alpha = 0.25$ (Lin et al., 2017). For the **Class-Balanced Cross-Entropy**, we applied effective number reweighting with $\beta = 0.9999$ (Cui et al., 2019). And for the **Class-Balanced Focal Loss**, we set $\beta = 0.9999$ and $\gamma = 2$ (Cui et al., 2019).

### A.3.2 COMPUTE RESOURCES

We trained all models using the PyTorch Image Models (Timm) framework or compatible extensions for specialized architectures. Training was performed using `torchrun` with distributed data parallelism across 4 NVIDIA V100 GPUs, most of the time and few models were trained on Google Cloud Platform using 16 NVIDIA V100 GPUs.

### A.4 ADDITIONAL RESULTS

### A.4.1 CLOSED-SET EVALUATION AND OPEN-SET RECOGNITION

We implemented a standard open-set recognition (OSR) protocol similar to (Vaze et al., 2022; Wang et al., 2024) on the cross-source split (Train-S4 / Test-S3&S4). Specifically, we held out 8 coral classes as unknown (e.g., *Agaricia*, *Pavona*), training only on the remaining 30 known classes. At test time, evaluation was performed on a mixed set containing both known (32,173 samples) and unknown (1,867 samples) classes. We followed a stratified hold-out protocol commonly used in OSR literature by excluding low-frequency classes and selecting unknowns with $\geq 100$ test-sample support for sound evaluation. Following (Vaze et al., 2022), we note that a classifier's closed-set

accuracy correlates strongly with its open-set detection ability. Accordingly, we use Max Logit Score (MLS) scoring as a competitive baseline, which their experiments show performs on par with specialized OSR methods. Consistent with (Wang et al., 2024), we evaluate multiple scoring rules: 1) Max Softmax Probability (MSP), 2) Max Logit Score (MLS), 3) Energy Score (Liu et al., 2020), and observe in Table 6 that magnitude-sensitive approaches like MLS and Energy yield the best performance, reinforcing their finding that such scoring functions generalize across both OSR and OOD tasks.

Table 6: **Open-set Recognition Results.** AUROC (higher is better) and FPR@95%TPR (lower is better) for different post hoc scoring methods.

| Method | AUROC ↑ | FPR@95%TPR ↓ |
|--------|---------|--------------|
| MSP    | 0.8169  | 0.6067       |
| MLS    | **0.8361** | **0.5820** |
| Energy | 0.1616  | 0.9902       |

### A.4.2 AUGMENTATION ABLATION RESULTS

While this work does not introduce new domain adaptation methods, our cross-domain experiments leveraged strong domain adaptation techniques, particularly through diverse data augmentation strategies. Specifically, we used a combination of RandAugment, color jitter, horizontal flipping, Mixup, CutMix, and random erasing (reprob = 0.1). These augmentations are widely used to improve generalization in the presence of domain shift due to variations in lighting, turbidity, and reef structure (Cubuk et al., 2020; Yun et al., 2019). To validate the contribution of these augmentations, we conducted an ablation study on the cross-source benchmark (Train-S4 / Test-S3&S4) using ViT-B (Dosovitskiy et al., 2021). The results in Table 7 demonstrate that using all augmentations yields the highest performance, confirming their collective benefit. Disabling any single augmentation (e.g., RandAug, ColorJitter, Random Erase) results in a performance drop, showing that each plays a role in enhancing robustness. Training without augmentation or with only a single augmentation significantly underperforms, validating that augmentation functions as an effective domain adaptation mechanism in our setup.

Table 7: **Effect of Data Augmentation Strategies.** Macro Recall and accuracy under different augmentation settings.

| Augmentation Setting | Macro Recall (%) | Accuracy (%) |
|----------------------|------------------|--------------|
| all-augmentation     | **50.49**        | 81.08        |
| no-randaug           | 49.47            | 80.78        |
| only-randerase       | 47.35            | 80.01        |
| no-randerase         | 50.44            | **81.36**    |
| no-augmentation      | 48.13            | 80.81        |
| only-hflip           | 46.93            | 79.80        |
| no-mixup             | 49.52            | 80.48        |
| no-cutmix            | 49.44            | 80.64        |
| only-mixup           | 46.04            | 79.40        |
| only-colorjitter     | 46.71            | 80.06        |
| no-hflip             | 49.84            | 80.96        |
| only-randaug         | 49.04            | 80.39        |
| no-colorjitter       | 49.91            | 80.53        |
| only-cutmix          | 48.29            | 80.12        |

### A.4.3 PER CLASS ANALYSIS OF VIT (MAE-PRETRAINED)

Since ViT (MAE-pretrained) fine-tuned on Train-S3 (evaluated on Test-S1) achieved high performance on the challenging cross-source benchmark, we provide a detailed per-class recall analysis for it on the (Train-S3 / Test-S3&S4) setup in Figure 10. Many of the misclassifications observed can be attributed to a combination of factors, primarily the low number of training images available for certain genera, in addition to morphological similarity and limitations related to image quality. Genera such as *Diploria*, *Meandrina*, and *Dendrogyra* were underrepresented in the training dataset,

which likely contributed significantly to their poor classification performance. More generally, when viewed in planar images, morphologically similar genera such as *Gardineroseris* and *Goniastrea*, or *Colpophyllia* and either *Platygyra* or *Pseudodiploria*, exhibit meandroid or sub-meandroid corallite structures that can appear nearly indistinguishable, especially for underwater images. Likewise, genera including *Acanthastrea*, *Dipsastrea*, and *Favites* show compact, polygonal corallites with prominent septa, leading to overlapping visual characters. Misclassifications between *Echinopora* and *Montipora*, or between *Echinopora* and *Porites*, may arise from comparable surface textures and encrusting to branching growth forms that become difficult to discriminate in images not focusing on the polyps. Similarly, genera such as *Siderastrea*, *Leptastrea*, *Pavona*, and *Agaricia* often have small, densely packed corallites and granular textures that challenge image-based classification. In pairs such as *Plesiastrea* and *Cyphastrea* or *Astreopora* and *Montipora*, the difficulty lies in distinguishing fine-scale textures and inconspicuous corallite structures. Finally, the confusion between *Goniastrea* and *Platygyra* represents a known case of convergent brain coral morphology, which can be difficult to resolve without high-resolution analysis on the skeleton.

In Figure 10, we present the top-1 error percentage across all errors for each class. For a more detailed view of per-class performance, Figure 11 shows the normalized confusion matrix of the model's predictions.

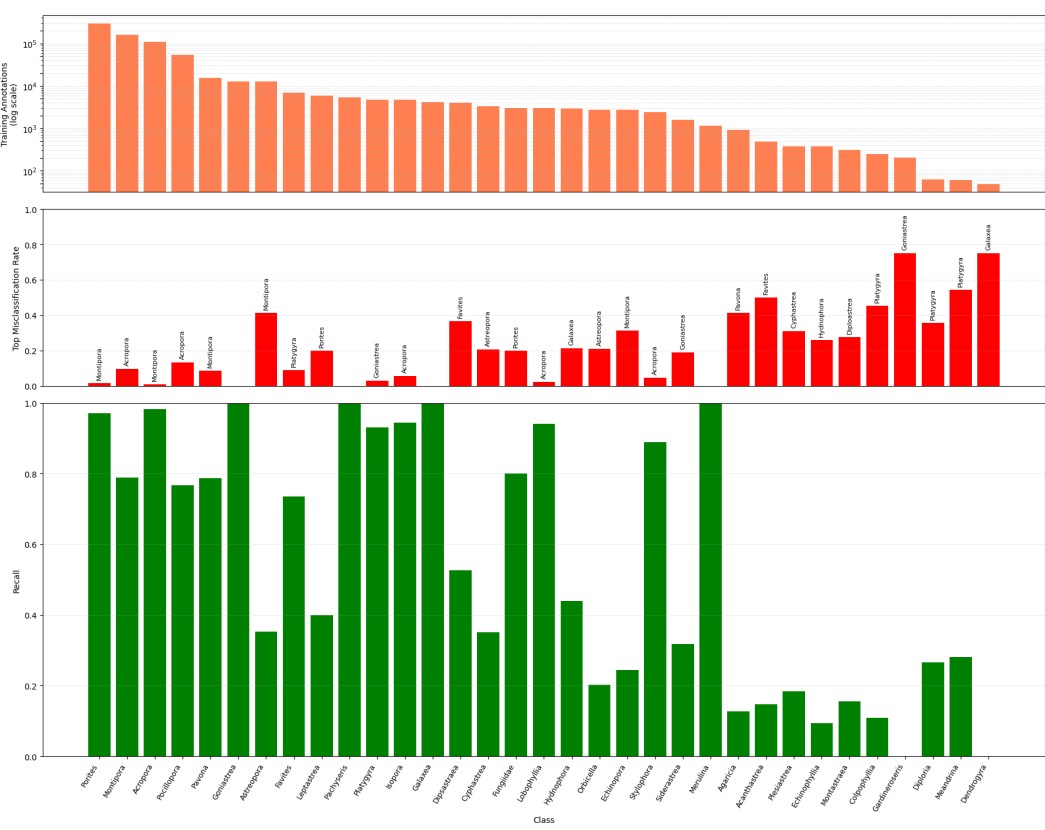

Figure 10: Per-class analysis of the ViT (MAE pretrained) model trained on the cross-source *Train-S3* and evaluated on the *Test-S3&S4*. Classes are ordered (left → right) by the number of training samples, shown in the **top** panel (log scale). The **middle** panel reports the fraction of all errors attributable to the single most frequent mis-label (top-1 misclassification rate). The **bottom** panel shows the per-class recall.

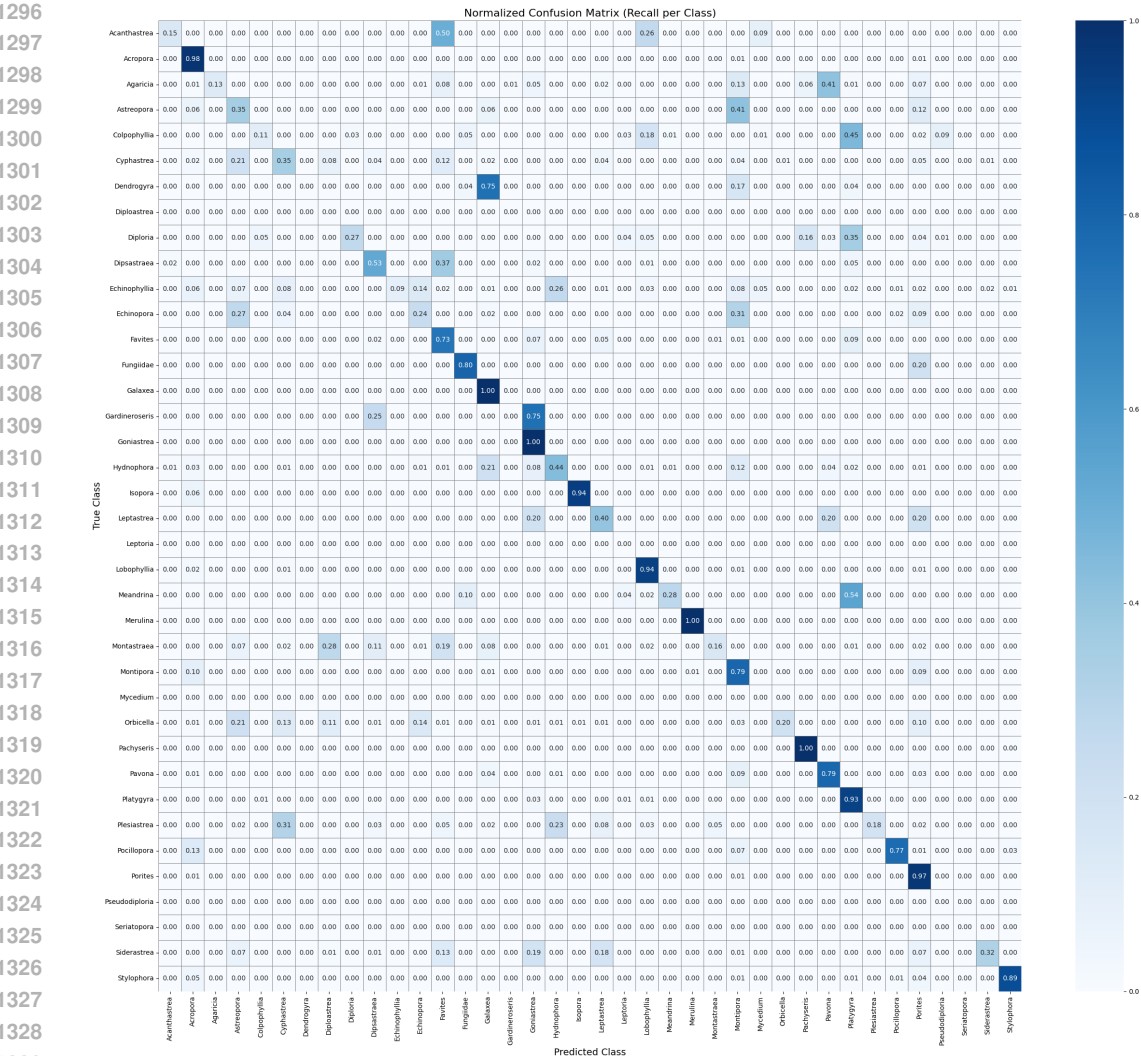

Figure 11: Normalized confusion matrix for the ViT MAE-pretrained model fine-tuned on the cross-source Train-S3 split and evaluated on the Test-S3&S4 set. The matrix includes all predicted classes (38 out of 39 possible labels in the Train-S3 set). Classes with no ground-truth samples in Test-S3&S4 (i.e., not part of its 33-label set) have zero values across their corresponding rows.

### A.4.4 OVERALL ACCURACY

Table 8 reports the overall accuracy of each model, measured as the micro-average recall across all classes. This metric captures the global classification performance but is inherently biased toward more abundant classes in both the training and test sets. The models shown correspond to the same experiments reported in Table 3 and Table 11 (updated BioCLIP-FT results).

### A.4.5 PRECISION AND F1 SCORE

Table 9 reports the F1 score of each model, calculated as the macro-average F1 across all classes. This metric provides a complementary view to the macro recall reported in Table 3. However, because some training classes may not appear in the test set, we cannot fully capture all false positives. As a result, both the precision values reported in Table 10 and the macro F1 scores may vary depending on the specific test classes available. In contrast, macro recall—used as our primary metric—remains consistent and unaffected by the absence of certain test classes.

Table 8: Hard coral classification **Micro Recall** under different train/test settings. Each model is evaluated using the micro-average recall, representing overall accuracy.

| | Train/Test split | | | | | |
|---|---|---|---|---|---|---|
| Model | within-source Train-S1 / Test-S1 | within-source Train-S2 / Test-S2 | cross-source Train-S3 / Test-S3 & S4 | cross-source Train-S4 / Test-S3 & S4 | cross-source Train-S3 / Test-W | cross-source Train-S4 / Test-W |
| EfficientNet Tan & Le (2019) | 97.24 | 96.53 | 80.79 | 81.15 | 93.56 | 86.90 |
| ResNet He et al. (2016) | 93.71 | 93.53 | 80.83 | 80.79 | 89.74 | 84.65 |
| ConvNext Liu et al. (2022) | 96.89 | 98.63 | 80.86 | 81.08 | 92.12 | 83.99 |
| ViT Dosovitskiy et al. (2021) | 97.27 | 98.05 | 81.08 | 81.02 | 93.75 | 87.95 |
| ViT (scratch) Dosovitskiy et al. (2021) | 86.38 | 89.27 | 80.72 | 80.90 | 92.92 | 87.51 |
| BEiT Bao et al. (2022) | 96.35 | 96.77 | 81.46 | 80.80 | 92.48 | 88.03 |
| DeiT Touvron et al. (2021) | 96.99 | 97.20 | 81.21 | 81.70 | 92.72 | 87.37 |
| Swin Liu et al. (2021) | 96.96 | 96.38 | 81.23 | 81.30 | 92.08 | 83.19 |
| ViT Dosovitskiy et al. (2021) (MAE He et al. (2022) pretrained) | 97.71 | 96.51 | 84.05 | 83.09 | 94.32 | 81.74 |
| BioCLIP-FT Stevens et al. (2023b) | 89.39 | 93.54 | 66.14 | 70.03 | 79.40 | 71.08 |
| Number of test classes | 39 | 39 | 33 | 33 | 12 | 12 |
| Number of test samples | 40,881 | 23,043 | 34,040 | 34,040 | 4,606 | 4,606 |

Table 9: Hard corals classification **Macro F1 Score** under different train/test settings.

| | Train/Test split | | | | | |
|---|---|---|---|---|---|---|
| Model | within-source Train-S1 / Test-S1 | within-source Train-S2 / Test-S2 | cross-source Train-S3 / Test-S3 & S4 | cross-source Train-S4 / Test-S3 & S4 | cross-source Train-S3 / Test-W | cross-source Train-S4 / Test-W |
| EfficientNet Tan & Le (2019) | 76.44 | 81.41 | 51.58 | 47.60 | 68.30 | 63.13 |
| ResNet He et al. (2016) | 64.96 | 65.14 | 44.69 | 33.21 | 64.00 | 54.81 |
| ConvNext Liu et al. (2022) | 75.33 | 80.37 | 47.81 | 51.02 | 67.98 | 64.12 |
| ViT Dosovitskiy et al. (2021) | 73.60 | 76.93 | 53.06 | 49.66 | 68.65 | 64.41 |
| ViT (scratch) Dosovitskiy et al. (2021) | 68.73 | 73.41 | 45.88 | 49.73 | 66.21 | 64.19 |
| BEiT Bao et al. (2022) | 71.12 | 72.14 | 50.38 | 37.21 | 64.77 | 62.54 |
| DeiT Touvron et al. (2021) | 72.68 | 75.23 | 50.20 | 47.14 | 64.09 | 64.34 |
| Swin Liu et al. (2021) | 75.41 | 81.39 | 50.13 | 48.07 | 65.56 | 62.62 |
| ViT Dosovitskiy et al. (2021) (MAE He et al. (2022) pretrained) | 77.36 | 75.33 | 57.57 | 51.77 | 70.22 | 59.80 |
| BioCLIP-FT Stevens et al. (2023b) | 65.16 | 69.06 | 66.14 | 70.03 | 64.99 | 59.73 |
| Number of test classes | 39 | 39 | 33 | 33 | 12 | 12 |
| Number of test samples | 40,881 | 23,043 | 34,040 | 34,040 | 4,606 | 4,606 |

Table 10: Hard corals classification **Macro Precision** under different train/test settings.

| | Train/Test split | | | | | |
|---|---|---|---|---|---|---|
| Model | within-source Train-S1 / Test-S1 | within-source Train-S2 / Test-S2 | cross-source Train-S3 / Test-S3 & S4 | cross-source Train-S4 / Test-S3 & S4 | cross-source Train-S3 / Test-W | cross-source Train-S4 / Test-W |
| EfficientNet Tan & Le (2019) | 74.66 | 80.48 | 55.73 | 55.33 | 67.32 | 61.72 |
| ResNet He et al. (2016) | 66.04 | 65.82 | 54.31 | 39.73 | 63.86 | 58.96 |
| ConvNext Liu et al. (2022) | 73.21 | 78.04 | 55.11 | 55.71 | 66.81 | 60.61 |
| ViT Dosovitskiy et al. (2021) | 71.80 | 75.00 | 55.93 | 55.66 | 67.62 | 62.72 |
| ViT (scratch) Dosovitskiy et al. (2021) | 63.23 | 66.57 | 54.57 | 55.47 | 66.25 | 62.36 |
| BEiT Bao et al. (2022) | 70.08 | 70.70 | 56.35 | 44.09 | 65.29 | 62.77 |
| DeiT Touvron et al. (2021) | 71.08 | 73.19 | 56.05 | 55.57 | 64.99 | 62.17 |
| Swin Liu et al. (2021) | 73.25 | 80.17 | 56.01 | 55.60 | 65.49 | 60.08 |
| ViT Dosovitskiy et al. (2021) (MAE He et al. (2022) pretrained) | 74.92 | 73.10 | 59.00 | 57.47 | 68.76 | 58.33 |
| BioCLIP-FT Stevens et al. (2023b) | 57.43 | 76.33 | 79.40 | 71.08 | 60.59 | 56.45 |
| Number of test classes | 39 | 39 | 33 | 33 | 12 | 12 |
| Number of test samples | 40,881 | 23,043 | 34,040 | 34,040 | 4,606 | 4,606 |

Table 11: Hard corals classification **Macro Recall** of **BioCLIP** under different train/test settings.

| | Train/Test split | | | | | |
|---|---|---|---|---|---|---|
| Model | within-source Train-S1 / Test-S1 | within-source Train-S2 / Test-S2 | cross-source Train-S3 / Test-S3 & S4 | cross-source Train-S4 / Test-S3 & S4 | cross-source Train-S3 / Test-W | cross-source Train-S4 / Test-W |
| BioCLIP-FT Stevens et al. (2023b) | 75.29 | 84.06 | 45.31 | 42.93 | 70.07 | 63.42 |

## A.5 Additional Qualitative Examples

This section presents qualitative examples illustrating model predictions and annotations from the ReefNet benchmarks. Figures 12 and 13 highlight predictions from the Test-S3 & S4 splits and the Al-Wajh dataset, respectively, offering insights into model performance across diverse hard coral categories.

In Figure 12, predictions from a cross-source model trained on Train-S4 demonstrate the model's ability to classify a range of coral genera such as *Porites*, *Acropora*, and *Montipora*. Ground-truth (GT) labels are displayed alongside model predictions and confidence scores, with correct

classifications shown in green and misclassifications in red. For instance, while the model achieves high precision on *Acropora* and *Porites*, it occasionally misclassifies *Porites* as *Montipora*, reflecting the difficulty of distinguishing morphologically similar coral types.

Figure 13 presents examples from the Al-Wajh dataset, which poses additional region-specific challenges. This dataset includes coral genera such as *Stylophora* and *Favia*. Notably, consistent confusion between *Porites* and *Montipora* underscores the difficulty of separating genera with overlapping morphological features. Nevertheless, the model demonstrates high confidence in classifying visually distinctive classes like *Goniastrea*.

These qualitative examples emphasize the inherent complexity of coral reef imagery, driven by factors such as morphological similarity, environmental variability (e.g., water clarity and lighting conditions), the presence of multiple benthic organisms in the same patch, and fuzzy class boundaries.

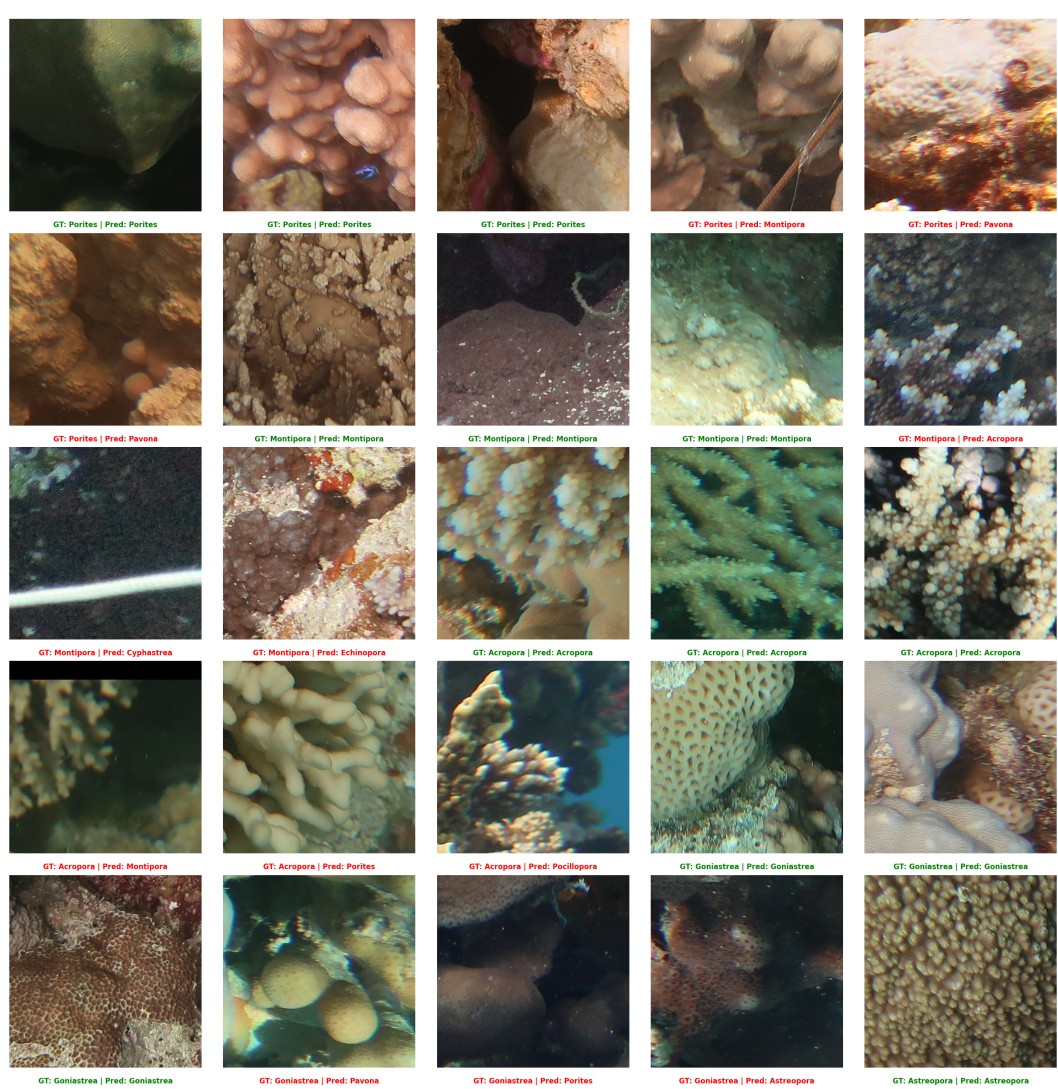

Figure 12: Qualitative examples from the ReefNet Cross-Source Test-S3. The model shown is a ViT finetuned using MAE pretraining. GT: Ground truth label; Pred: Model prediction.

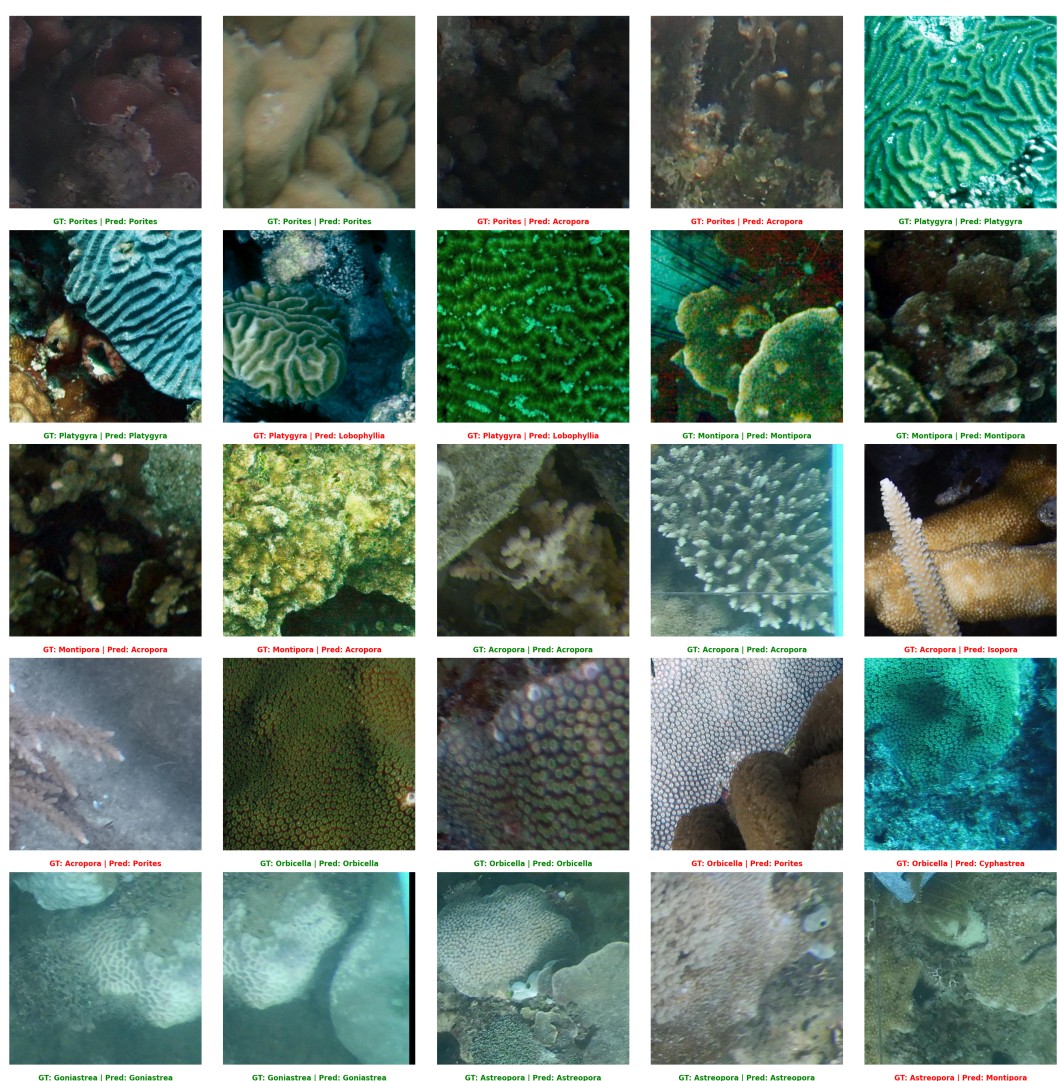

Figure 13: Qualitative examples from the Al-Wajh dataset. The model shown is a ViT finetuned using MAE pretraining. GT: Ground truth label; Pred: Model prediction.

## A.6 Text Description Generation and performance for Zero-Shot models

We use two approaches to generate descriptions for various coral genera, both using GPT-4o. 1) We directly prompt GPT-4o to generate a description for a coral genus with the prompt presented in Figure 14. 2) In the second approach, we provide context information about the genus extracted from domain experts' chosen books Wallace (1999); Veron (2000a;b) and online sources by allowing an agent to scrape results from Google search. The second approach was done using an agentic pipeline built using CrewAI. The system included one agent per book, one agent for online sources, and a final agent to take the information from the previous agents and produce the final output.

From Table 12, it can be seen that across the three vision language models trained on general data from the internet, OpenCLip performs the best. BioCLIP clearly outperformed all the other models across all the splits, supporting the idea that its pretrained data from TreeOfLife-10M Stevens et al. (2023c), which contains coral information, supports its performance.

For the Within-source Test-S2 and Cross-source Test-S3&S4 splits, the benchmark includes up to 39 unique hard coral genera. Each genus is paired with two textual descriptions, referred to as

```
Produce a sentence, without any additional context, that
doesn't exceed 30-40 words describing the coral class
'coral_class' with a focus on including as many descriptive
adjectives as possible.  The sentence does not need to be
grammatically perfect.  Be as descriptive of its physical
appearance as possible (color, texture, shape, etc).
```

Figure 14: ChatGPT prompt used to generate class descriptions.

Table 12: **Average Rank of the Ground Truth Class for Each Model Across Data Splits**. All the Vision Language Models use generated textual descriptions for the Coral genus.

| Split | CLIP | SigLIP | OpenCLIP | BioCLIP |
|---|---|---|---|---|
| within-source | 15.96 | 15.03 | 14.12 | 10.33 |
| within-source (QC) | 18.00 | 17.32 | 16.52 | 10.40 |
| cross-source | 17.29 | 16.16 | 15.29 | 9.13 |
| cross-source (QC) | 16.42 | 15.48 | 14.67 | 8.89 |

"GPT" and "Book," which were generated using our described pipeline. Although the pipeline can produce additional descriptions, we restrict our experiments to one description per genus in both the "Qwen-GPT" and "Qwen-Book" settings. This design isolates inference-time performance without introducing further training, thereby demonstrating how such descriptions enhance the zero-shot classification ability of Qwen2.5-VL.

## A.7 LIMITATIONS

While the ReefNet dataset marks a significant advancement in providing standardized, taxonomically fine-grained annotations for coral reef imagery, several limitations should be considered when training models or developing applications based on this dataset:

- **Dynamic Taxonomy:** Coral taxonomy is an evolving field of science, and taxonomic classifications are subject to revision over time. While the dataset reflects the most current understanding at the time of compilation, future changes in taxonomy may render some annotations outdated. Users are encouraged to leverage the provided AphialDs to verify the latest accepted taxonomy through the World Register of Marine Species (WoRMS) (Board, 2024).

- **Patch-Based Annotations:** ReefNet annotations are exclusively patch-based, which may not fully capture the spatial variability and ecological context of coral reefs (e.g., Figure 4). Patch-based models, while valuable, may be less accurate than models leveraging more comprehensive semantic segmentation. Although recent large-scale segmentation datasets, such as CoralVOS and CoralSCOP (Zheng et al., 2024; Ziqiang et al., 2023), offer valuable insights, they lack fine-grained taxonomic data. ReefNet complements these datasets by providing extensive taxonomically detailed annotations, filling a critical gap for future integration with segmentation-based approaches.

## A.8 CONTRIBUTOR ATTRIBUTION AND ETHICAL DATA USE.

To ensure transparency, traceability, and proper credit to original data providers, we compiled a comprehensive list of all 85 CoralNet sources included in ReefNet, along with their corresponding contributors and institutional affiliations. This attribution acknowledges the global community of researchers and practitioners who have made their data publicly available through CoralNet, often as part of long-term ecological monitoring efforts. The information was curated through a combination of CoralNet metadata, institutional websites, and direct outreach to contributors. In line with principles of responsible data stewardship, we have preserved all original attribution metadata and encourage future users of ReefNet to do the same.

## A.9 RESOURCES AVAILIBITY

The taxonomically mapped annotations, ReefNet metadata, trained models, and the Al-Wajh Lagoon image collection we collected will be made publicly available on the Hugging Face platform. For the corresponding CoralNet imagery, we will provide a list of source image URLs and a script to download the data directly from CoralNet, thereby leaving control over the images with the original owners. The code will be released on GitHub.

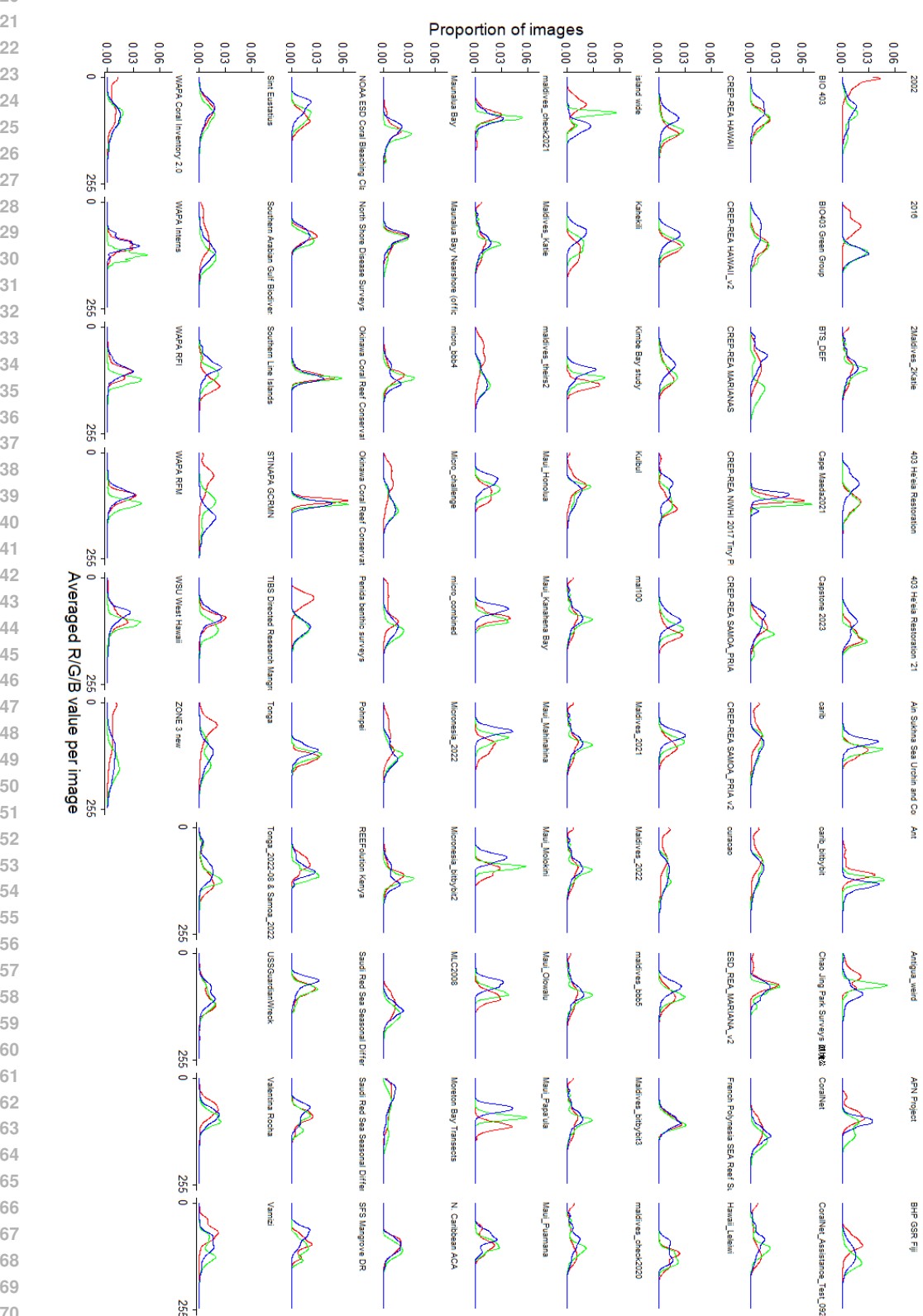

Figure 15: Density plots per CoralNet source of the average Red, Green, and Blue values of each image.

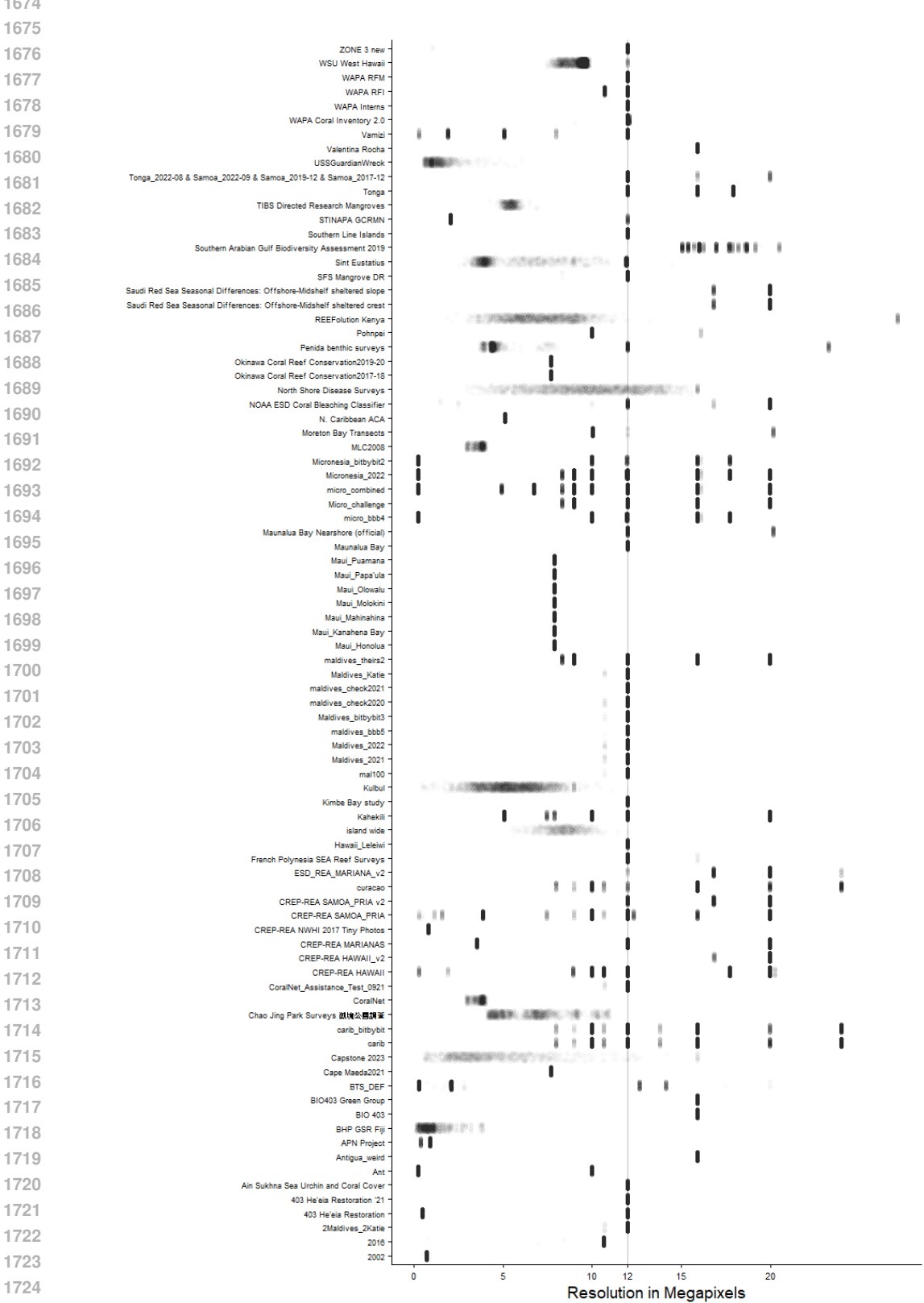

Figure 16: Scatter plot showing the resolution of each annotated image per source.

Table 13: Contributors and affiliations for the 85 CoralNet sources included in the ReefNet dataset. This list is based on publicly available information from the CoralNet platform, supplemented by our in-depth online research into each source and outreach to their original contributors.

| CoralNet source | Listed contributors | Listed affiliations |
|---|---|---|
| 2002 | Zuhairah Dindar | University of New South Wales |
| 2016 | Zuhairah Dindar | University of New South Wales |
| 2Maldives_2Katie | Katie Lubarsky, Hugh Runyan | University of California San Diego |
| 403 He'eia Restoration | Cynthia Hunter, Dorian Brunzelle, Kaitlyn Jacobs, Alana Minato, Cody Powers, Connor Antonis, Devon Stapleton, Dylan Rich, Jeany Robledo, Jacob Nygaard, Jordan Pounds-Crihfield, Jacquelyn Simpson, Kevin Christensen, Kaitlin Hooper, Madeline Payne, Renee Wold, Shayna Arakaki, Zachary Clark | Hawaii Institute of Marine Biology, University of Hawaii at Manoa |
| 403 He'eia Restoration '21 | Cynthia Hunter, Dorian Brunzelle, Kaitlyn Jacobs, Keisha Bahr, Brooklyn Bennett, Marisa Bhao-Intr, Brittany Kernodle, Corey Ling, Dan Zhuo, Desiree Shaw, Johann Vollrath, Kenzie Vierra, Lena Marinkovich, Lauryn Pisciotto, Mellisa Gajardo, Madeleine Perez, Sophia Hanscom, Samantha Thomas, Toranosuke Degawa, Zada Boyce-Quentin | Hawaii Institute of Marine Biology, University of Hawaii at Manoa |
| Ain Sukhna Sea Urchin and Coral Cover | Omar Attum | Indiana University Southeast |
| Ant | Hugh Runyan | Scripps Institution of Oceanography, University of California San Diego |
| Antigua_weird | Hugh Runyan | Scripps Institution of Oceanography, University of California San Diego |
| APN Project | Roslizawati Ab Lah, Kirsten Benkendorff, Zoe White | University of Malaysia Terengganu, Southern Cross University |
| BHP GSR Fiji | John Stratford, Jason Lynch | University College London, Zoological Society London |
| BIO 403 | Morgan Guadagnoli, Ana Velasquez, Eliza Beckwith, Haley Weis, Isabella Davila, Jamie Mazurski, Rachel Bagnas, Terra Stevens | University of Hawaii at Manoa |
| BIO403 Green Group | Morgan Guadagnoli, Ana Velasquez, Eliza Beckwith, Haley Weis, Terra Stevens | University of Hawaii at Manoa |
| BTS_DEF | Henrique dos Santos, Igor Cruz, Joao Ferreira, Ian Vinicio | Universidade Federal da Bahia |
| Cape Maeda2021 | Tomofumi Nagata | Okinawa Environment Science Center |
| Capstone 2023 | Emily Ogawa, David Hyrenbach, Kristina Bechthold, Leslie Rosa, Ivy Haxo | Hawaii Pacific University |
| carib | Hugh Runyan, Nathaniel Hanna Holloway | Scripps Institution of Oceanography, University of California San Diego |
| carib_bitbybit | Hugh Runyan | Scripps Institution of Oceanography, University of California San Diego |
| Chao Jing Park Surveys | Emma Chen, Shinya Shikina, Kaixiang Yang, Chu Yu Ling, Joey Hsia, Yu-Chund Chuan, Tzu-Cheng Lin, Cheng Yin Chu, Wen Teng Huang, Yuenyi Leung | National Taiwan Ocean University |
| CoralNet | Dong Li | Zhejiang University |
| CoralNet_Assistance_Test_092 | Hugh Runyan, Ceiba Becker, Esmaralda Alcantar, Nicole Pedersen | Scripps Institution of Oceanography, University of California San Diego |

*Continued on the next page*

| CoralNet source | Listed contributors | Listed affiliations |
|---|---|---|
| CREP-REA HAWAII NOAA Pacific Islands Fisheries Science Center, Ecosystem Sciences Division (2018b) | Annette DesRochers, Andrew Gray, Brett Schumacher, Bernardo Vargas Angel, Courtney Couch, Ivor Williams, Jonathan Charendoff, Morgan Winston Pomeroy, Paula Misa, Tom Oliver, Troy Kanemura, Ari Halperin, Chelsie Counsell, Colt Davis, Isabelle Basden, Jon Ehrenberg, Kerry Reardon, Mia Lamirand, Roseanna Lee | National Oceanographic and Atmospheric Administration, Pacific Islands Fisheries Science Center |
| CREP-REA MARIANAS NOAA Pacific Islands Fisheries Science Center, Ecosystem Sciences Division (2018a) | Annette DesRochers, Andrew Gray, Bernardo Vargas Angel, Courtney Couch, Jonathan Charendoff, Morgan Winston Pomeroy, Paula Misa, Tom Oliver, Troy Kanemura, Kaylyn McCoy, Kevin Lino, Marie Ferguson, Mia Lamirand, Nalani Kito-Ho, Winter Jimenez, Brett Schumacher | National Oceanographic and Atmospheric Administration, Pacific Islands Fisheries Science Center |
| CREP-REA NWHI 2017 Tiny Photos NOAA Pacific Islands Fisheries Science Center, Ecosystem Sciences Division (2018b) | Andrew Gray, Bernardo Vargas Angel, Courtney Couch, Jon Ehrenberg | National Oceanographic and Atmospheric Administration, Pacific Islands Fisheries Science Center |
| CREP-REA SAMOA/PRIA NOAA Pacific Islands Fisheries Science Center, Ecosystem Sciences Division (2018c) | Annette DesRochers, Andrew Gray, Brett Schumacher, Bernardo Vargas Angel, Courtney Couch, Ivor Williams, Jonathan Charendoff, Morgan Winston Pomeroy, Paula Misa, Tom Oliver, Troy Kanemura, Isabelle Basden, Mia Lamirand, Andrew Shantz, Brittany Huntington, Corinne Amir, Hatsue Bailey, Marie Ferguson, Mollie Asbury, Nalani Kito-Ho, Winter Jiminez, Helen Ford, Nicole Kamalu | National Oceanographic and Atmospheric Administration, Pacific Islands Fisheries Science Center |
| curacao | Hugh Runyan | Scripps Institution of Oceanography, University of California San Diego |
| ESD_REA HAWAII_v2 NOAA Pacific Islands Fisheries Science Center, Ecosystem Sciences Division (2018b) | Annette DesRochers, Bernardo Vargas Angel, Courtney Couch, Jonathan Charendoff, Morgan Winston Pomeroy, Tom Oliver, Isabelle Basden, Jon Ehrenberg, Hatsue Bailey, John Morris, Mia Larimand, Paula Misa | National Oceanographic and Atmospheric Administration, Pacific Islands Fisheries Science Center |
| ESD_REA_MARIANA_v2 NOAA Pacific Islands Fisheries Science Center, Ecosystem Sciences Division (2018a) | Andrew Gray, Bernardo Vargas Angel, Courtney Couch, Jonathan Charendoff, Kaylyn McCoy, Tom Oliver, Ari Halperin, Mia Lamirand, Hatsue Bailey, Nicolas Osborn, Jon Ehrenberg | National Oceanographic and Atmospheric Administration, Pacific Islands Fisheries Science Center |
| ESD_REA_SAMOA_PRIA_v2 NOAA Pacific Islands Fisheries Science Center, Ecosystem Sciences Division (2018c) | Andrew Gray, Bernardo Vargas Angel, Courtney Couch, Jonathan Charendoff, Morgan Winston Pomeroy, Paula Misa, Ari Halperin, Isabelle Basden, Mia Lamirand, Hatsue Bailey, Nicolas Osborn, Tom Oliver | National Oceanographic and Atmospheric Administration, Pacific Islands Fisheries Science Center |
| French Polynesia SEA Reef Surveys | Elliott Bates | International Master of Science in Marine Biological Resourses, Sea Education Association |
| Hawaii_Leleiwi | Russel Sparks, Devon Aguiar | Department of Land and Natural Resources - Aquatics |
| island wide | Daniela Escontrela, Elena Turner | University of Hawaii |
| Kahekili | Bernardo Vargas Angel, Ivor Williams, Andrew Gray,Tye Kindinger, Mia Lamirand, | National Oceanographic and Atmospheric Administration, Pacific Islands Fisheries Science Center |
| Kimbe Bay study | Alice Williams, Kitty Watts | University of Bristol |
| Kulbul | Ben Murphy, Caitlin Younis, Hannah Kish, Azri Saparwan, Justin Bovery-Spencer, Tarquin Singleton | GBR Biology |

*Continued on the next page*

| CoralNet source | Listed contributors | Listed affiliations |
| --- | --- | --- |
| mal100 | Hugh Runyan | Scripps Institution of Oceanography, University of California San Diego |
| Maldives_2021 | Katie Lubarsky, Hugh Runyan, Anupama Sethuraman, Ceiba Becker, Jamie Pettengell | Scripps Institution of Oceanography, University of California San Diego |
| Maldives_2022 | Hugh Runyan | Scripps Institution of Oceanography, University of California San Diego |
| Maldives_bitbybit3 | Hugh Runyan | Scripps Institution of Oceanography, University of California San Diego |
| maldives_bbb5 | Hugh Runyan | Scripps Institution of Oceanography, University of California San Diego |
| maldives_check2020 | Hugh Runyan | Scripps Institution of Oceanography, University of California San Diego |
| maldives_check2021 | Hugh Runyan | Scripps Institution of Oceanography, University of California San Diego |
| Maldives_Katie | Hugh Runyan, Katie Lubarsky | Scripps Institution of Oceanography, University of California San Diego |
| maldives_theirs2 | Hugh Runyan | Scripps Institution of Oceanography, University of California San Diego |
| Maui_Honolua | Russel Sparks | Department of Land and Natural Resources - Aquatics |
| Maui_Kanahena Bay | Russel Sparks | Department of Land and Natural Resources - Aquatics |
| Maui_Mahinahina | Russel Sparks | Department of Land and Natural Resources - Aquatics |
| Maui_Molokini | Russel Sparks | Department of Land and Natural Resources - Aquatics |
| Maui_Olowalu | Russel Sparks, Tatiana Martinez | Department of Land and Natural Resources - Aquatics |
| Maui_Papa'ula | Russel Sparks, Tatiana Martinez | Department of Land and Natural Resources - Aquatics |
| Maui_Puamana | Russel Sparks | Department of Land and Natural Resources - Aquatics |
| Maunalua Bay | Paula Moehlenkamp | Univeristy of Hawaii |
| Maunalua Bay Nearshore (official) | Pamela Weiant, Alexandria Barkman | Malama Maunalua |
| micro_bbb4 | Hugh Runyan | Scripps Institution of Oceanography, University of California San Diego |
| Micro_challenge | Hugh Runyan, Katie Lubarsky | Scripps Institution of Oceanography, University of California San Diego |
| micro_combined | Hugh Runyan, Katie Lubarsky, Chris Sullivan, Ahmyia Cacapit, Charles Hambley, Isa Bersamin, Sarah Romero | Scripps Institution of Oceanography, University of California San Diego |

| CoralNet source | Listed contributors | Listed affiliations |
|---|---|---|
| Micronesia_2022 | Hugh Runyan, Katie Lubarsky, Chris Sullivan | Scripps Institution of Oceanography, University of California San Diego |
| Micronesia_bitbybit2 | Hugh Runyan | Scripps Institution of Oceanography, University of California San Diego |
| MLC2008 | Dong Li | Zhejiang University |
| Moreton Bay Transects | Joshua Wirth, Gal Eyal | University of Queensland |
| N. Caribbean ACA | Alexandra Ordonez Alvarez, Brianna Bambic, Myles Phillips, Bernadette Charpentier | National Geographic, Queensland University, Wildlife Conservation Society, University of Ottowa |
| NOAA ESD Coral Bleaching Classifier Ehrenberg et al. (2022) | Courtney Couch, Jonathan Charendoff, Morgan Winston Pomeroy, Tom Oliver, Jon Ehrenberg | National Oceanographic and Atmospheric Administration, Pacific Islands Fisheries Science Center |
| North Shore Disease Surveys | Julianna Renzi, Maddie Cunningham | University of California Santa Barbara |
| Okinawa Coral Reef Conservation2017-18 | Tomofumi Nagata, Eiji Yamakawa | Okinawa Environment Science Center |
| Okinawa Coral Reef Conservation2019-20 | Tomofumi Nagata | Okinawa Environment Science Center |
| Penida benthic surveys | Pascal Sebastian, Rinaldi Gotama | Indo Ocean Project |
| Pohnpei | Hugh Runyan | Scripps Institution of Oceanography, University of California San Diego |
| REEFolution Kenya | Ewout G. Knoester, Anniek Vos, Bulisa Masiga, Jowan van Lente, Luc Visser, Mercy Zawadi Katana, Omar F. Yusuf | Wageningen University and Research, REEFolution Trust |
| Saudi Red Sea Seasonal Differences: Offshore-Midshelf sheltered crest | Clara Nuber, Matt Tietbohl, Karla Gonzalez | King Abdullah University of Science and Technology |
| Saudi Red Sea Seasonal Differences: Offshore-Midshelf sheltered slope | Clara Nuber, Matt Tietbohl, Karla Gonzalez | King Abdullah University of Science and Technology |
| SFS Mangrove DR | Max Vierling, Samantha Krausse, Toni Trinh | School for Field Studies |
| Sint Eustatius | Myrsini Lymperaki | Univeristy of Amsterdam |
| Southern Arabian Gulf Biodiversity Assessment 2019 | Jeneen Hadj-Hammou, John Burt, Rita Bento | New York University Abu Dhabi |
| Southern Line Islands | Nicole Pedersen, Samantha Clements | Scripps Institution of Oceanography, University of California San Diego |
| STINAPA GCRMN | Caren Eckrich, Tessa Haanskorf, Angelica Verschragen | STIchting NAtionale PArken Bonaire, Wageningen University and Research, University of Amsterdam |
| TIBS Directed Research Mangroves | Emma Greenberg, Jenna Shea, Rachel Schneider | School for Field Studies |
| Tonga | Patrick Smallhorn-West, Lucy Southworth | James Cook University |
| Tonga_2022-08 & Samoa_2022-09 & Samoa_2019-12 & Samoa_2017-12NOAA Pacific Islands Fisheries Science Center, Ecosystem Sciences Division (2018c) | Chris Sullivan, Katie Lubarsky, Gloria Mariño-Briceño, Hannah Gower, Kylie Yogi, Phi Lang | Scripps Institution of Oceanography, University of California San Diego |
| USSGuardianWreck | Catherine Kim, Ben Neal, Dominic Bryant | Univeristy of Queensland |
| Valentina Rocha | Valentina Rocha, Emily Esplandiu | University of Miami |

*Continued on the next page*

| CoralNet source | Listed contributors | Listed affiliations |
|---|---|---|
| Vamizi | Marques da Silva Isabel, Erwan Sol, Felix Domadoma | Center for Research and Environmental Conservation - Lurio University |
| WAPA Coral Inventory 2.0 | David Burdick, Colin Lock, Melissa Vaccarino | National Park Service |
| WAPA Interns | Ashton Williams, Marisa Agarwal, Andrew O'Connor, Christina Kilkeary, Emma Vaughn, Erin Mullins, Katherine Tangney, Malvika Shrimali, Michelle Diminuco, Motusaga Vaeoso, Natalie Scott, Nicholas Burgos, Philippe Astier, Ryan Stanley, Sarah Yokota, Serena Butler, Ashley Swafford, Xavier Quinata | National Park Service |
| WAPA RFI | Anneke Padmos, Julia Padilla, Ronja Steinbach, Tim Clark, Terence Dela Cruz, Eliza Frances Manglona | National Park Service |
| WAPA RFM | Anneke Padmos, Julia Padilla, Ronja Steinbach, Tim Clark, Terence Dela Cruz | National Park Service |
| WSU West Hawaii | Brian Tissot, Molly Bogeberg | Washington State University |
| ZONE 3 new | Jamila Hassan, Sulemani Mohamed | Wildlife Conservation Society |

