# OpenReview forum: "ReefNet: A Large-Scale, Taxonomically Enriched Dataset and Benchmark for Hard Coral Classification"
_ICLR.cc/2026/Conference — Submitted to ICLR 2026_

### Official Review · Reviewer_yu4R · 2025-10-28

**Soundness:** 2
**Presentation:** 2
**Contribution:** 2
**Rating:** 2
**Confidence:** 5

**Summary:**

The paper introduces ReefNet, a coral reef image dataset that unifies data from 76 CoralNet sources and an additional Red Sea subset, containing about 920K genus-level annotations mapped to the World Register of Marine Species (WoRMS). It establishes two standardized benchmarks, i.e., within-source (in-domain) and cross-source (out-of-domain), to evaluate model generalization under domain shift and class imbalance conditions.  The paper’s contributions include dataset standardization and expert verification, the introduction of a Red Sea subset, and empirical evaluation of supervised and zero-shot models on these benchmarks.

**Strengths:**

**1.** The paper presents a coral dataset that integrates multiple data sources under a unified taxonomy aligned with the WoRMS, improving data consistency across heterogeneous origins.

**2.** It establishes two standardized evaluation settings, within-source and cross-source, to assess model performance under both in-domain and domain-shift conditions in coral classification tasks.

**3.** The introduction of a Red Sea subset from an underrepresented region broadens the dataset’s geographic diversity and may support further research on generalization across biogeographic contexts.

**Weaknesses:**

**1. Incremental contribution within the Datasets and Benchmarks scope:** ReefNet mainly aggregates existing CoralNet sources and applies genus-level standardization using WoRMS taxonomy. The inclusion of a Red Sea subset broadens coverage but does not represent a conceptual or methodological advance compared with prior marine datasets such as BenthicNet or CoralNet.

**2. Use of coarse, patch-based annotations:** As the paper itself acknowledges, ReefNet relies on sparse, patch-based point annotations, which are not precise for coral classification because a single patch can contain multiple coral species. This limits the dataset’s ability to support fine-grained visual understanding. In contrast, datasets such as CoralSCOP and CoralVOS provide mask-based annotations that capture pixel-level structure and are better suited for detailed ecological analysis.

**3. Lack of analytical depth in benchmarking:** The experimental results confirm well-known trends, i.e., strong within-domain accuracy but severe degradation under domain shift and weak zero-shot generalization, without providing deeper insight into underlying causes or dataset-specific challenges.

**4. Unclear advantage over existing resources:** Table 1 shows that some prior datasets already offer species-level labels and comparable or larger image volumes. The paper does not convincingly demonstrate how ReefNet enables new research directions beyond those datasets. Although standardized taxonomy and expert re-verification is valuable for ecological data management, these are engineering-level enhancements rather than conceptual advances.

**Questions:**

**1.** Table 1 shows that BenthicNet already provides a larger image volume, species-level annotations, and WoRMS mapping, which suggests a higher level of taxonomic detail and scale than ReefNet. Could the authors clearly explain what advantages ReefNet offers over BenthicNet?

**2.** It would be interesting to see the future work incorporating additional modalities, such as depth data, 3D reconstructions, or temporal sequences, to enhance the dataset’s relevance for representation learning.

**3.** Could the authors clarify the planned release timeline and accessibility details of ReefNet, including availability of preprocessing or benchmark scripts?

---

### Official Review · Reviewer_Es6T · 2025-10-29

**Soundness:** 1
**Presentation:** 1
**Contribution:** 1
**Rating:** 0
**Confidence:** 4

**Summary:**

The paper introduces ReefNet, a large-scale, WoRMS-mapped dataset and benchmark for genus-level hard-coral classification, aggregating ≈925k point labels from 76 CoralNet sources plus a new Red Sea site. It defines within-source and cross-source splits and evaluates a suite of supervised backbones and zero-shot VLMs/MLLMs, reporting strong in-source results but sharp performance drops under domain shift. Despite useful data aggregation, the submission reads like an engineering/curation report with limited novelty, shallow evaluation, and unclear positioning vs. existing large marine datasets. The scientific contribution falls short of ICLR’s high bar for novelty, rigor, and insight. I recommend strong reject.

**Strengths:**

+ The curation scale and WoRMS alignment (genus-level) are valuable to the community, and the within- vs cross-source protocols are reasonable to study domain shift.
+ The paper documents a quality-control pipeline (expert review → filtering) and provides detailed split summaries (S1–S4, Test-W).
+ Baseline runs show the severity of cross-site degradation; the result tables/ablations (e.g., focal vs class-balanced loss) may guide practitioners.

**Weaknesses:**

+ My main concern is the insufficient novelty for ICLR main track. The paper is fundamentally a dataset/benchmark release with conventional baselines (ResNet/EfficientNet/ViTs, CLIP variants) and straightforward loss tweaks; there is no new learning method, no new DG/robustness technique, and limited analysis beyond reporting macro-recall drops. ICLR typically expects algorithmic or theoretical advances or significantly deeper analytical insights than shown here.
+ Table 1 acknowledges BenthicNet and other resources at global scale and even notes WoRMS support, undermining the “first/most comprehensive” narrative. The manuscript does not convincingly differentiate ReefNet beyond re-mapping CoralNet labels and adding one Red Sea site. A rigorous head-to-head benchmark against BenthicNet subsets or standardized taxonomies is missing.
+ Initial expert agreement is ~73%, later filtered to 78–92% by removing weak source/genus pairs. This heavy pruning raises questions about selection bias and external validity of the “high-confidence” subset (S2/S4), yet the paper leans on these filtered splits to claim reliability. A careful analysis of what is discarded and how this affects class/geography distributions is not provided.
+ Macro recall is the primary metric; confidence intervals/seed variation are not reported per model/split in Table 3, and there is little analysis of calibration, per-class long-tail behavior, or OOD diagnostics beyond a brief note and a loss ablation.
+ No modern domain generalization baselines (e.g., strong DG/DA methods, test-time adaptation, distributionally robust training) appear; conclusions that “models struggle” under shift are unsurprising without a competitive DG suite.
+ The dataset is hierarchical, but the benchmark does not evaluate hierarchical metrics (e.g., hierarchical precision/recall, taxonomic distance penalties), missing an obvious and informative lens.
+ I also find the zero-shot/MLLM setup as weak scientifically. The paper prompts Qwen-VL and even uses GPT-4-generated or book-summarized genus descriptions to boost zero-shot numbers. This ad-hoc retrieval/LLM prompting feels methodologically fragile, and the takeaways (BioCLIP best; others low) are not surprising given pretraining corpora. As “results,” they do not elevate the scientific contribution.
+ I am also very concerned about the licensing/ethics and provenance, as the proofs are thin. ReefNet aggregates from 76 CoralNet sources and scanned books for text; while the paper says releases will comply with licenses, it does not detail per-source license heterogeneity, derivative-work permissions for text, long-term hosting, or update/versioning of a taxonomy that evolves (WoRMS). These are critical for a flagship benchmark.

**Questions:**

+ Can you substantively differentiate ReefNet from BenthicNet and other existing resources via controlled, head-to-head experiments (same taxonomic mapping, same backbones, same splits) rather than narrative comparisons?
+ The zero-shot section relies on GPT-generated/summarized text to prompt MLLMs. What is the scientific takeaway for ICLR beyond confirming pretraining coverage effects? Can you replace this with a reproducible retrieval method and ablate prompt sources?
+ Please provide licensing and provenance tables per source (image/text), explicit redistribution permissions, and a versioning policy for future WoRMS taxonomic updates.

**Details Of Ethics Concerns:**

My big ethics concern is the dataset’s licensing and provenance: ReefNet aggregates imagery from 76 CoralNet sources and external texts (incl. book scans/GPT-generated summaries) without a clear, per-source breakdown of redistribution rights, derivative-work permissions, or long-term hosting and versioning policies for a taxonomy that changes over time. This is compounded by heavy expert-filtering that alters class/site distributions without quantifying what was removed, potentially introducing selection bias while still claiming “high-confidence” labels. The zero-shot section’s reliance on GPT-produced or book-summarized genus descriptions further raises reproducibility and copyright risks, and may import unexamined biases from those sources.

---

### Official Review · Reviewer_oa4H · 2025-10-30

**Soundness:** 2
**Presentation:** 2
**Contribution:** 2
**Rating:** 4
**Confidence:** 4

**Summary:**

The paper introduces ReefNet, a large-scale coral reef image dataset aimed at advancing coral classification. The dataset integrates imagery from 76 curated CoralNet sources and an additional Red Sea collection from the Al-Wajh Lagoon. It includes approximately 925k genus-level hard coral point annotations, each mapped to the WoRMS and verified under the supervision of domain experts. Two benchmark settings are proposed: a within-source benchmark for assessing in-distribution performance and a cross-source benchmark for measuring out-of-distribution generalization across distinct reef sites. Experimental results show that models achieve strong within-source performance but experience substantial performance degradation across sources, underscoring the continued difficulty of achieving domain generalization in underwater image classification.

**Strengths:**

1) Despite the dataset’s considerable ecological and practical value, its contribution to machine learning and computer vision research remains limited. The work primarily centers on dataset construction and benchmarking rather than presenting new learning methods, optimization techniques, or analytical frameworks. For a high-impact venue such as ICLR, where methodological novelty is a key expectation, this focus on resource building may restrict the paper’s broader appeal.

2) Another strong aspect is the taxonomic precision and alignment with a recognized global standard (WoRMS). By offering genus-level labels rather than coarse taxonomic groups, the dataset provides a richer foundation for ecological research and biodiversity modeling.

**Weaknesses:**

1) While the dataset’s ecological value is undeniable, its contribution to computer vision or machine learning is less substantial. The paper focuses on resource creation and benchmarking rather than proposing new modeling techniques or addressing fundamental algorithmic questions. For a top-tier venue like ICLR, where methodological novelty is often prioritized, this may limit the work’s appeal.

2) A major concern is the dataset’s continued reliance on **point-based annotations**, which substantially limit spatial expressiveness. While such annotations are efficient to collect, they fail to capture essential structural cues, including object boundaries, surface textures, and morphological context, that are crucial for dense prediction tasks like semantic segmentation, coverage estimation, or 3D reconstruction. Although ReefNet offers a wider range of semantic labels than existing datasets such as MosaicsUCSD, CoralSCOP, and Coralscapes, which provide **dense mask annotations**, ReefNet’s point annotations inherently constrain downstream applications  and appear less aligned with current computer vision practices emphasizing spatially rich data. The paper would benefit from a clearer and more compelling justification for maintaining a point-based annotation scheme rather than transitioning toward mask- or region-level representations.

3) Furthermore, the findings from the benchmark experiments lack conceptual novelty within the broader field of machine learning. The observation that models experience **performance degradation under cross-source or out-of-distribution scenarios largely reaffirms well-established phenomena**. The paper would be significantly strengthened by a deeper and more domain-specific analysis of why coral classification presents unique challenges compared to the general classification. For instance, discussing how disordered coral morphologies, overlapping growth forms, and environmental factors influence coral visual appearance and finally impact the classification accuracy could yield valuable biological and computational insights. Exploring these aspects would elevate the work beyond a dataset paper, offering a richer understanding of domain shift in ecological imagery.

**Questions:**

My main question regarding this paper is about the limited methodology, superficial analysis and missing explanation of the degraded performance.

---

### Official Review · Reviewer_E5Ys · 2025-10-31

**Soundness:** 3
**Presentation:** 3
**Contribution:** 2
**Rating:** 4
**Confidence:** 4

**Summary:**

This paper uses a subset of CoralNet and one set of newly collected images to produce a dataset of hard coral point annotations. They convened a team of expert coral annotators to review and refine existing labels. The dataset was split a variety of ways to test in- vs. out-of-distribution performance of a number of fine-tuned CNNs, transformers, and BioCLIP. Performance degraded in the out-of-distribution situation. The authors additionally did some zero-shot tests, noting that BioCLIP did best.

**Strengths:**

- The authors come up with some interesting out-of-distribution test splits based on where data is collected. The following tests reveal potentially informative structure of the dataset.
- The quality of the dataset is quite high. The authors went to a great deal of effort to develop a thorough domain expert review process with a team of coral experts. That information is largely found in the supplement and could be fore-fronted in the main text.
- The writing is reasonably clear, though the naming conventions for the out-of-distribution tests splits are somewhat confusing.
- The results, especially regarding label consistency of publicly available coral datasets, is an important issue for marine conservation.

**Weaknesses:**

Degradation of model performance in out-of-distribution situations is a well-documented issue, especially in data poor domains. The point is made in papers like the [WILDS benchmark](https://proceedings.mlr.press/v139/koh21a) and the BioCLIP paper itself. The primary contribution of this work seems to be more domain specific, rather than revealing anything novel from a learning perspective.

The dataset is, for the moment, relatively small with ~300k images containing ~925k point annotations of 44 classes. The authors seem to intend to grow the dataset which may make some of these results more compelling.

**Questions:**

- Can you elaborate on how much overlap there is between the BioCLIP label space and that of ReefNet?
- Can you contextualize the zero-shot performance of the VLMs on ReefNet relative to other out-of-distribution datasets? This need not require new experiments, just some literature review.
- How do non-domain specific VLMs perform when fine-tuned on the various data splits?
- How representative of coral diversity are the 44 genera in ReefNet? This is almost certainly a complicated question, but some indication of what the complete space looks like will be helpful to contextualize how ReefNet fits in.
- Please expand on the statement at line 482 that ReefNet-trained models can be used to pre-label common genera. What constitutes a 'common genera'? Is there a heuristic to define that in terms of label frequency across regions?

---

### Meta-Review · Area_Chair_zYkA · 2026-01-07

**Summary:**

The paper introduces ReefNet, a benchmark dataset of coral reef images sourced from CoralNet (a similar previously existing coral dataset) + additional images curated by the authors. From the AI perspective, the contribution is a reasonably challenging benchmark for visual fine-grained classification.

The paper received uniformly negative reviews. While the reviewers appreciated the (potential) ecological contributions, the sentiment was that the dataset itself was just a mildly expanded and label-remapped version of existing datasets, and didn't represent a methodological advance; moreoever, the findings on out-of-distribution generalization performance of vision-language models were all mostly echoing well-established trends.

**Reviewer Concerns:**

N/A, since there was no rebuttal.

**Reviewer Scores:**

N/A, since there was no rebuttal.

---

### Decision · Program_Chairs · 2026-01-26

Reject